# Live imaging of alveologenesis in precision-cut lung slices reveals dynamic epithelial cell behaviour

Khondoker M. Akram [1], Laura L. Yates[1], Róisín Mongey [1], Stephen Rothery[2], David C.A. Gaboriau [2], Jeremy Sanderson[3], Matthew Hind [1,4], Mark Griffiths[1,5] & Charlotte H. Dean [1,3]

Damage to alveoli, the gas-exchanging region of the lungs, is a component of many chronic and acute lung diseases. In addition, insufficient generation of alveoli results in broncho-pulmonary dysplasia, a disease of prematurity. Therefore visualising the process of alveolar development (alveologenesis) is critical for our understanding of lung homeostasis and for the development of treatments to repair and regenerate lung tissue. Here we show live alveologenesis, using long-term, time-lapse imaging of precision-cut lung slices. We reveal that during this process, epithelial cells are highly mobile and we identify specific cell behaviours that contribute to alveologenesis: cell clustering, hollowing and cell extension. Using the cytoskeleton inhibitors blebbistatin and cytochalasin D, we show that cell migration is a key driver of alveologenesis. This study reveals important novel information about lung biology and provides a new system in which to manipulate alveologenesis genetically and pharmacologically.

[1] National Heart and Lung Institute, Imperial College London, London, UK. [2] Facility for Imaging by Light Microscopy, NHLI, Faculty of Medicine, Imperial College London, London, UK. [3] MRC Harwell Institute, Harwell Campus, Oxfordshire, UK. [4] National Institute for Health Research (NIHR) Respiratory Biomedical Research Unit at the Royal Brompton & Harefield NHS Foundation Trust and Imperial College, London, UK. [5] Peri-Operative Medicine Department, St Bartholomew's Hospital, London, UK. Correspondence and requests for materials should be addressed to C.H.D. (email: c.dean@imperial.ac.uk)

The primary function of the lungs is gas exchange and the site for this is the alveoli[1,2]. The gas exchange surface maximises surface area whilst minimising the barrier to diffusion from the airspace to the circulation. It is comprised of two thin cellular layers of alveolar epithelium and capillary endothelium[3]. There is a significant need to understand the mechanisms of alveolar formation because a number of neonatal and infant diseases, including bronchopulmonary dysplasia (BPD) and pulmonary hypoplasia, involve insufficient generation of alveoli[4,5]. In addition, damage to the alveolar region is a component of several chronic adult lung diseases such as chronic obstructive pulmonary disease (COPD) and idiopathic pulmonary fibrosis (IPF) and a cause of acute respiratory failure in pneumonia and acute respiratory distress syndrome (ARDS). Currently, there is almost a complete absence of disease-modifying treatments for these very common conditions. The pivotal role of alveoli in lung function and disease, has led to an increasing focus on alveolar biology[6–8].

The structure of mature alveoli has been elucidated primarily from 2-dimensional static images, however, their formation is not well understood, since this requires a way of visualising the process in real-time, something that is difficult to do in an organ that lies deep within the body and which takes place almost entirely after birth in humans and completely after birth in mouse. In contrast, detailed knowledge of airway generation, which occurs in utero, prior to alveolarisation, has been gained from both static and ex vivo real-time imaging experiments because counterintuitively, mouse embryonic lungs are both practically and experimentally more accessible[9–11]. X-ray tomography and imaging of lung vibratome sections combined with genetic labelling have added to our knowledge of alveologenesis by generating static, 3-dimensional pictures of this process at different time-points[12,13].

A recent study by Li et al. used both ex vivo and in vivo live imaging to study the sacculation stage of lung development, immediately prior to alveologenesis[14], but these techniques are not suitable for imaging postnatal lungs[15].

In mice, sacculation begins at embryonic day (E) 17.5, lasting until the first few days of postnatal life[1]. During this stage, the primitive air sacs form from the distal airways and distal tip epithelial cells begin to express markers indicative of their differentiation into mature type I (ATI) and type II (ATII) alveolar epithelial cells, such as podoplanin and pro-surfactant protein C (SP-C) respectively. Subsequent to this, alveolarisation begins shortly after birth. The most active, 'bulk' alveolarisation phase lasts until postnatal day (P) 14 and the majority of alveoli are formed by P21[16,17]. Largely based on inference from static images, it is thought that alveoli form by repeated septation events that sub-divide primitive airspaces thereby increasing the surface area for gas exchange[12,18]. Cell proliferation is considered to play a key role in alveologenesis, with many publications showing that it increases at the onset of bulk alveolarisation around P4 and then rapidly declines towards the end of this developmental phase. However, the methods used to measure proliferation and the cell types analysed vary widely between studies, as does the level of proliferation reported[19–22]. Cell migration is also believed to be important in alveologenesis, particularly for septation to occur, but it has not been possible to confirm the relative contributions of migration and proliferation until now.

Precision cut lung slices (PCLS) contain intact alveoli, rather than monolayers of one or two cells types (co-cultures). Crucially, in contrast to organoids, cell types are present in the same ratios and with the same cell–cell and cell–matrix interactions as in vivo. Human and mouse ex-vivo PCLS are increasingly used to study disparate aspects of lung biology[23–25]. Recent publications have shown that alveologenesis can be tracked in early postnatal PCLS, using static snapshot images[19]. In addition, time-lapse imaging of PCLS combined with immunolabelling has revealed dynamic interactions of mesenchymal cells and macrophages with the extracellular matrix in adult normal and fibrotic mouse lungs, as well as in PCLS of human lungs[26]. These studies raised the possibility that alveologenesis could be visualised in real-time using imaging of early postnatal mouse lung PCLS.

Here we reveal the morphological mechanisms of alveologenesis in real-time, using widefield microscopy and image deconvolution[27]. Using this technique, we show that early postnatal epithelial cells are highly dynamic. We identify three distinct behaviours adopted by epithelial cells that facilitate alveologenesis: cell clustering; hollowing and cell extension. Using Precision Cut Lung Slice imaging (PCLSi), combined with pharmacological inhibitors of the actin-myosin cytoskeleton, we show that epithelial cell migration plays a dominant role in early alveologenesis.

## Results

**Establishing real-time imaging of PCLS**. To investigate alveologenesis, PCLS (300 μm thick) were prepared from postnatal mice and cultured using a modified protocol described in ref. [19] (Fig. 1a). For live imaging experiments, PCLS were placed in a humidified chamber, attached to a widefield microscope, and images were captured every 15 min for up to 19 h (Fig. 1b). In parallel with live imaging, additional PCLS were cultured for up to 72 h to enable the capture of further parameters, including cell proliferation, mean linear intercept (Lm) and optimal doses, of cytoskeletal inhibitors (Fig. 1b). Serum-free DMEM (SF-DMEM) was superior to supplemented M199 media[19] for viability of PCLS up to 96 h in culture (Supplementary Figure 1a, b) and phenol-free media was used to aid fluorescent imaging (Supplementary Figure 1c). The viability of PCLS was determined by measuring metabolic activity, Live/Dead and caspase assays (Supplementary Figure 1a–c).

To track individual epithelial cells during alveologenesis, we looked for reagents to easily and selectively label these cells in the PCLS. EpCAM (also known as CD326), a cell surface antigen expressed on epithelial cell membranes, has been widely used as an epithelial cell marker[28–30]. We tested a FITC-conjugated EpCAM antibody and found this provided efficient labelling of live epithelial cells (Fig. 1c–e, g–i), without altering cell viability (Fig. 1f). Typical epithelial cell membrane labelling was observed in both alveolar (Fig. 1d) and bronchial epithelial cells (Fig. 1e) within PCLS and the efficiency with which EpCAM labelled epithelial cells was also confirmed by double immunostaining with EpCAM and pan-cytokeratin antibodies (96.98% of cytokeratin labelled cells were EpCAM positive, Supplementary Figure 2a–d, i). DNA conjugated to silicon rhodamine fluorophore (SiR-DNA) effectively labelled all cell nuclei within PCLS (Fig. 1c–e, g–i). Both of these live cell labels allowed individual cells to be tracked in the PCLS for up to 19 h without significant bleaching and did not alter cell viability during time-lapse imaging (Fig. 1f). Longer time-lapse experiments (up to 64 h) were also conducted, however, the tissue was not viable after approximately 21–24 h, compared to PCLS cultured under normal (non-time-lapse) conditions and therefore longer-term experiments were discontinued (Supplementary Movie 1A, B; Supplementary Figure 1e).

We confirmed the specificity of the EpCAM-FITC antibody by triple labelling P3 mouse PCLS with EpCAM-FITC, SP-C and SiR-DNA, the majority of EpCAM-labelled epithelial cells were also positively labelled with Sp-C (Fig. 1g, arrows). A small number of EpCAM positive cells in the lung parenchyma were Sp-C negative, indicating, as expected, that there were other

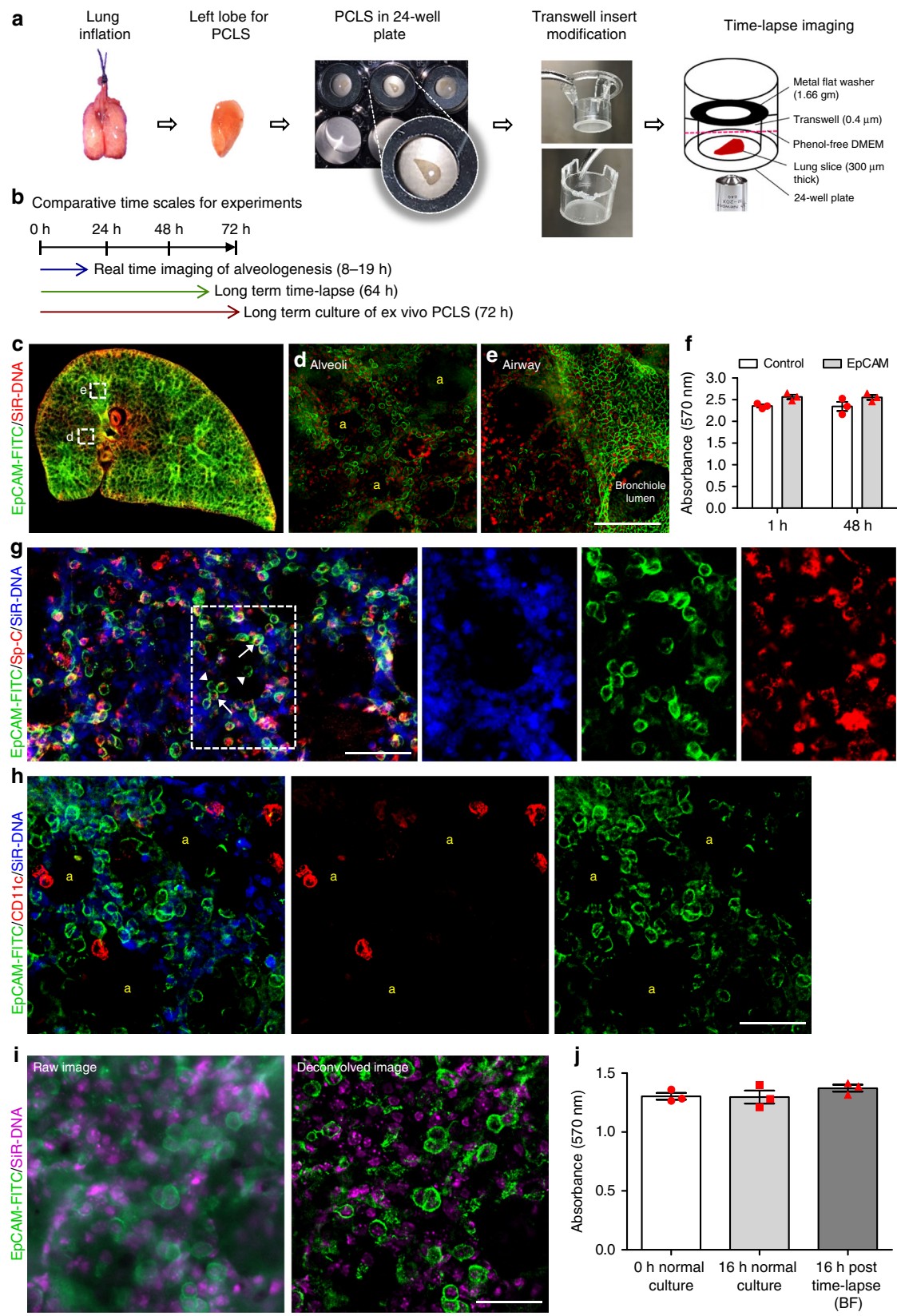

epithelial subtypes present e.g., ATI (Fig. 1g, arrowheads). Triple labelling with EpCAM-FITC, SiR-DNA and a live cell marker for macrophages, CD11c conjugated to PE (Phycoerythrin: CD11c-PE) showed there was no off-target uptake of EpCAM by macrophages (Fig. 1h; Supplementary Figure 1d). Labelling of

PCLS with isotype control fluorophores showed no non-specific staining, confirming the specificity of the antibodies used for live imaging (Supplementary Figure 8). We also observed a few Sp-C positive but EpCAM negative cells and further investigation revealed a small population of Sp-C positive cells that were also

**Fig. 1** Experimental protocol for time-lapse imaging of precision cut lung slices. **a** Lung harvesting, inflation, PCLS generation and imaging. **b** Comparative durations of ex vivo PCLS culture experiments. **c–e** Deconvolved widefield, single plane z-stack image of P3 PCLS. Epithelial cells were labelled with EpCAM-FITC antibody (green) and cell nuclei were labelled with SiR-DNA (red). Boxed areas show EpCAM-FITC +ve cells in alveolar (d) and airway (e) epithelial cells. Scale bar = 50 μm. **f** MTT cell viability assay comparing control and EpCAM-FITC/SiR-DNA labelled adult PCLS at 1 and 48 h. $n = 3$ independent experiments, with duplicate slices per group, per experiment. **g** Confocal images of P3 PCLS labelled with EpCAM-FITC, Sp-C (alveolar type 2 cell marker) and SiR-DNA. EpCAM-FITC (green)/SP-C (red) +ve epithelial cells (arrows) EpCAM-FITC (green)/ Sp-C (red) −ve cells (arrowheads). Nuclei were stained with SiR-DNA (blue). $n = 2$ independent experiments, with duplicate slices from one mouse per experiment. Scale bar = 50 μm. **h** Deconvolved widefield, single plane, z-stack image of P3 PCLS labelled with EpCAM-FITC (green)/CD11c-PE (red, macrophage marker)/SiR-DNA (blue). $n = 2$ independent experiments, with duplicate slices from one mouse per experiment. Scale bar = 50 μm. **i** Raw and deconvolved, widefield, single plane, z-stack images labelled with EpCAM-FITC (green)/SiR-DNA(magenta). Scale bar = 50 μm. **j** MTT cell viability assay on PCLS pre and post imaging and in normal culture conditions, $n = 3$ independent experiments using three separate mice, with duplicate slices per group per experiment. Yellow 'a' indicates alveolar airspaces. Error bars are defined as s.e.m

positive for CD11c (Supplementary Figure 2e–i). Therefore, EpCAM and SiR-DNA dual labelling was used for real-time imaging of epithelial cells during alveologenesis.

Confocal imaging through the entire 300 μm z-axis of PCLS, labelled with live/dead reagents, revealed the presence of some dead cells on and near the cut surfaces of the PCLS, but very few dead cells were observed deeper into the tissue (Supplementary Figure 3a–d). Therefore, to facilitate efficient capture of widefield time-lapse movies from healthy tissue, we took z-stacks through the middle portion of the PCLS only (50–60 μm for brightfield and 15–20 μm for fluorescence) (Supplementary Figure 3b). We captured data from four separate fields per slice and images were acquired every 15 mins for 19–21 h (Supplementary Figure 3a, b).

To reduce fluorophore bleaching during time-lapse acquisition, we opted to use widefield microscopy[31]. However, the raw images obtained by this method were hazy and unsuitable for individual cell observations due to the nature of widefield imaging, and light diffraction caused by sample thickness. Deconvolution uses an algorithm to eliminate the out-of-focus light and increase resolution, resulting in clearer sharper images[27,32]. We found that deconvolution of widefield z-stacks removed much of the out-of-focus light and improved image quality to a near-confocal standard, with the added benefit of acquisition speed (Fig. 1i). We, therefore, investigated applying iterative deconvolution processing to all time-lapse z-stack sequences and found that using this method, we could obtain movies in which individual cells could be visualised and tracked (Supplementary Movie 2a, b). We also confirmed that 16 h of time-lapse imaging had no significant effect on cell viability compared to normal cell culture (Fig. 1j, Supplementary Figure 4).

To confirm the timing of alveologenesis in C57BL/6J postnatal mice, lungs from P3, P7, P14 and adult mice were fixed, sectioned and stained with H&E prior to quantifying the mean linear intercept (Lm) and the number of airspaces. A smaller Lm value reflects a higher-density of septa and more alveoli. In our study, Lm was significantly lower in P7 compared with P3 (63.83 μm vs. 98.69 μm, respectively; $p < 0.001$), and P14 Lm was significantly lower than P7 ($p < 0.001$); however, there was no significant difference in Lm values between adult and P14 lungs (45.16 μm vs. 43.50 μm, respectively) (Fig. 2a–e). Comparison of Lm in PCLS taken from 6 individual P3 mice showed no significant variability in mice of the same age between experiments (Fig. 2f).

We also found the number of airspaces was 2.1-fold higher in P7 than that of P3 ($p < 0.001$); and significantly higher in P14 compared with P7 ($p < 0.001$). There was no significant difference between adult and P14 groups (513 airspaces/mm² vs. 460 airspaces/mm² respectively) (Fig. 2a–d, g).

To determine the levels of cell proliferation during alveologenesis, lungs were harvested from P3, P7, P14 and adult (6–8 weeks) mice, embedded and sectioned for immunostaining. Quantification of the percentage of Ki67 positive cells showed that

proliferation was higher at P7 than P3 (23.1% vs. 17.58%, respectively; $p < 0.05$;). However, the number of Ki67 positive cells dropped sharply at P14 compared with P7 (6.2% vs. 23.1%, respectively; $p < 0.001$;), and there was no significant difference between P14 and adult lungs (6.2% vs. 5.7%, respectively; Supplementary Figure 5a). These data correlated with published findings indicating that the bulk of alveologenesis is complete by P14. Specific quantification of EpCAM, Ki67 dual positive epithelial cells in PCLS during bulk alveologenesis, showed that in the ex-vivo PCLSi model, the mean total of proliferating cells was 16.1%(P3) and 15.8%(P7) however the level of epithelial cells proliferating accounted for approximately half of this, 9.8%(P3) and 8.8%(P7) (Supplementary Figure 5b).

**Epithelial cell motility during murine alveologenesis**. To quantify epithelial cell dynamics during alveologenesis, we compared epithelial cell motility in P3, P7, P14 and adult PCLS, labelled with EpCAM-FITC, during 8 h time-lapse movies. Movie sequences were subsequently analysed using Icy open source software[33] to track and quantify epithelial cell movement. Mean net cell migration (total linear migration of a cell from point 'A' to point 'B' in 8 h) was quantified and found to be significantly higher at P3 compared to P7 (Fig. 2h, i, $p < 0.001$; Supplementary Movie 3A, B). Net epithelial cell migration was also higher in P7 when compared with that of P14 (Fig. 2h, i; $p < 0.01$, Supplementary Movie 3B, 4A). In contrast, cell migration was minimal in P14 and adult lung (Fig. 2h, i, Supplementary Movie 4A, B) and there was no significant difference between these two groups (mean 1.52 μm vs. 1.82 μm, respectively). Further, data analysis revealed that a large proportion of epithelial cells were relatively sessile and migrated less than 2 μm at all age groups. However, at P3 a significant proportion of epithelial cells were much more motile; 6% of cells migrated between 6 and 13 μm. At P7, the proportion of these highly motile cells dropped to 1.9% of cells migrating between 6 and 9 μm ($p < 0.01$; Fig. 2j). The presence of this highly motile cell group was negligible in P14 and adult lung (Fig. 2j). These data indicated that of the time points tested, the highest levels of epithelial cell migration occurred in P3 PCLS. Therefore, PCLS from P3-P4 mice were used for subsequent time-lapse experiments.

**PCLSi identifies clustering and hollowing of epithelial cells.** Current understanding is that, lung alveologenesis occurs through repeated secondary septation events, which sub-divide existing airspaces to produce additional airspaces and increase the surface area. Using fluorescent PCLSi, we were able to capture this type of event, where an EpCAM positive secondary septum-like structure gradually protrudes further into an existing airspace to sub-divide it (Fig. 3a, b; Supplementary Movie 5A; red blocked arrows mark secondary septa, red dashed line indicates sub-dividing airspaces,

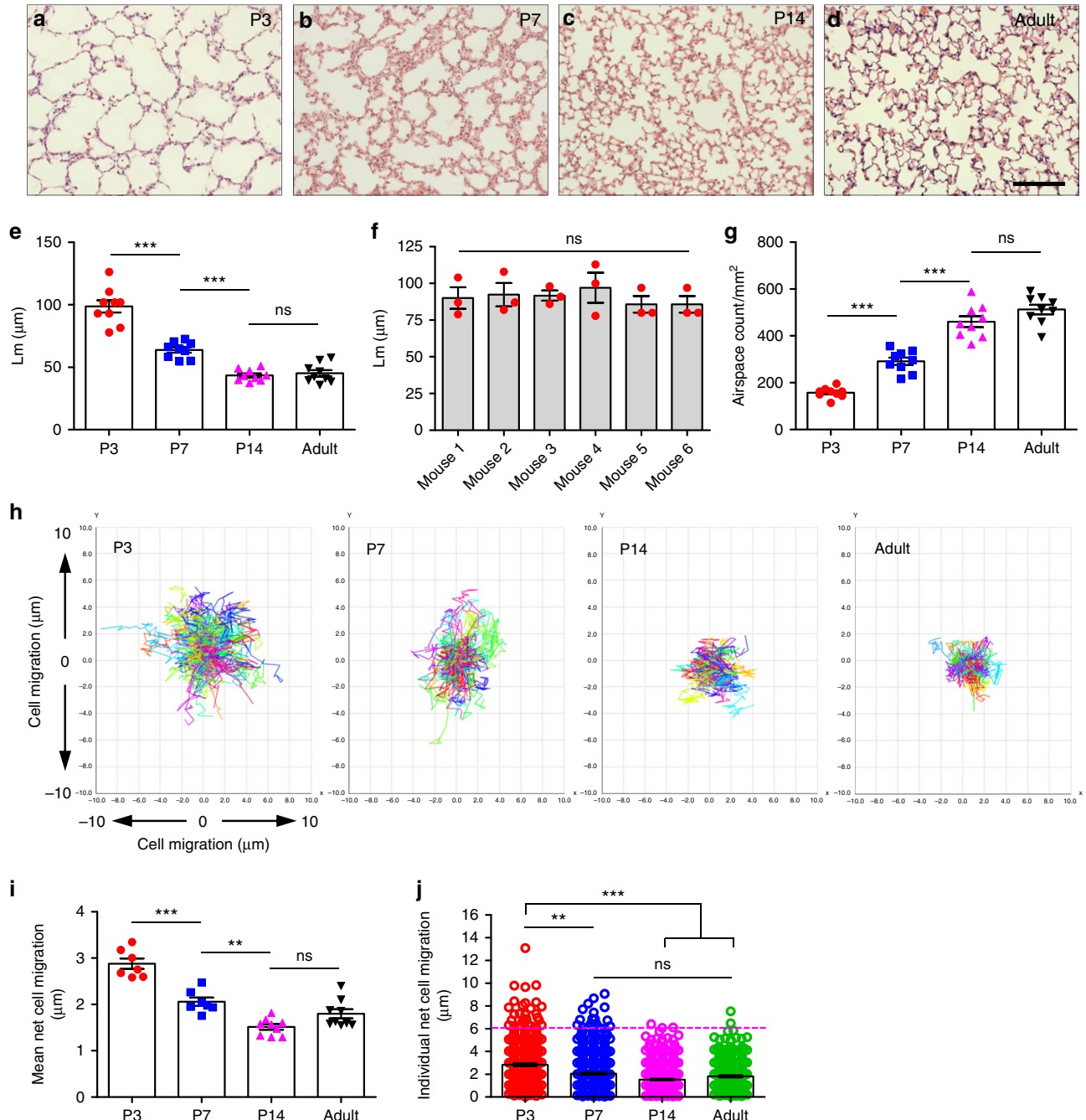

**Fig. 2** Quantification of alveologenesis and alveolar epithelial cell migration in postnatal mice. H&E staining of left lung sections from P3, P7, P14 and adult mice (**a–d**); n = 3 mice from each age group. Scale bar = 100 µm. Mean linear intercept (Lm) from P3, P7, P14 and adult mice (**e**); n = 3 separate mice from each age group, 3 H&E lung sections were quantified per mouse, ***p < 0.001; ns = not significant, one-way ANOVA with Tukey's post hoc test. Mean linear intercept in PCLS from P3 mice (**f**), n = 6 separate mice, 3 H&E lung sections were quantified per mouse; ns = not significant, one-way ANOVA with Tukey's post hoc test. Airspace count from P3, P7, P14 and adult lung sections (**g**); n = 3 separate mice from each age group, 3 H&E lung sections were quantified per mouse, ***p < 0.001; ns = not significant, one-way ANOVA with Tukey's post hoc test. Individual cell tracking over 8 h in P3, P7, P14 and adult PCLS, 70–160 cells were tracked per field (**h**). Mean net epithelial cell migration over 8 h in P3, P7, P14 and adult lungs (**i**); **p < 0.01, ***p < 0.001; ns = not significant, one-way ANOVA with Tukey's post hoc test, n = 3 mice for each age group, 2–3 fields were quantified from one slice per mouse, per experiment. Individual net cell migration in P3, P7, P14 and adult lungs over 8 h (each dot represents a single cell). P3, 607 cells, P7, 944 cells, P12, 950 cells and adult, 613 cells were tracked (**j**), n = 3 mice from each age group, 2–3 fields were quantified from one slice per mouse, per experiment, **p < 0.01, ***p < 0.001, ns = not significant. Error bars are defined as s.e.m

a1–a2, a3–a4). Details of septation can be seen in zoomed in movies of a1–a2 (Supplementary Movie 5B) and a3–a4 (Supplementary Movie 5Ci).

Fluorescent labelling of PCLS highlighted that many EpCAM positive epithelial cells were exceedingly dynamic.

Moreover, careful analysis of the movies revealed details about alveolar formation that could not previously be appreciated from still images. We frequently observed cell clustering, where multiple epithelial cells migrate towards a single spatial area to form a cluster (Fig. 3a, b; blue circles,

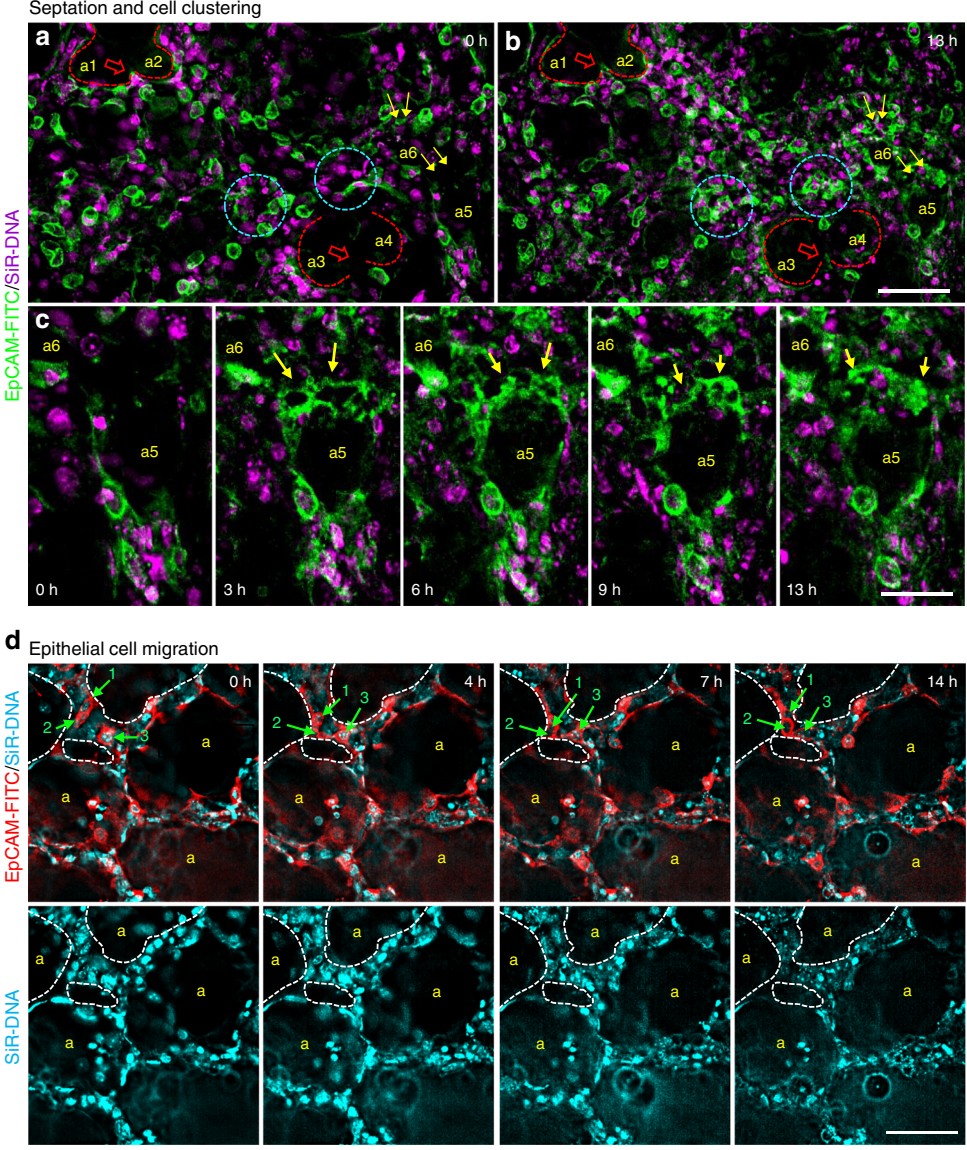

**Fig. 3** Septation, cell clustering and cell migration in PCLSi. Deconvolved widefield images of P3 PCLS from 13 h time-lapse video (Supplementary movie 5A and higher magnification 5B, 5Ci) show two examples of septation, i.e., extension of secondary septum into an airspace (dashed red line demarcates airspaces a1/a2 and a3/a4, red boxed arrows point to secondary septa (**a**, **b**). Epithelial cell clustering, a common feature noted during postnatal alveologenesis, is indicated by blue circles (**a**, **b**) (Supplementary movie 5A and higher magnification 5D). The images also show EpCAM-FITC (green) labelled alveolar epithelial cell migration (yellow arrows) towards existing airspaces (a5 and a6) (**a**–**c**). Enlarged still images from 13 h time-lapse video (Supplementary movie 5A higher magnification 5E at 0, 3, 6, 9 and 13 h, show epithelial cell migration (yellow arrows) towards airspace a5 (**c**). Nuclei were labelled with SiR-DNA (magenta). Scale bar = 50 μm (**a**, **b**), 25 μm (**c**). Deconvolved EpCAM-FITC (red)/SiR-DNA (cyan) labelled still images from 14 h time-lapse video (Supplementary movie 6) showing EpCAM positive epithelial cell migration (green arrows) towards alveolar airspace (white dashed lines outline airspaces) in P3 PCLS (**d**). Cells 1 and 2 migrate towards a small airspace (white dashed outline) within the interstitium and meet cell 3 (**d**, upper panel). Corresponding images labelled with SiR-DNA only are displayed in **d**, lower panel. Yellow 'a' indicates alveolar airspaces. Scale bar = 50 μm

Supplementary movies 5A, blue circles and zoomed in Supplementary movie 5D).

We also observed cells migrating to contribute to an existing alveolus. Here individual cells migrated from one area of the parenchyma towards a newly formed airspace, where they adopted a position adjacent to other epithelial cells around the perimeter of the airspace (Fig. 3a–c, a5 and a6, yellow arrows, Supplementary movie 5A, a5 and a6, yellow arrows and zoomed in Supplementary movie 5E). Similar behaviour was observed in a separate experiment, where epithelial cells from an adjacent area can be seen migrating towards the boundary of a small newly formed cell-free space and adhering to another epithelial cell at

that site (Fig. 3d, green arrows, cell 1 and 2 migrate towards airspace and meet cell 3; Supplementary movie 6).

Another event that we observed in the movies was hollowing. In this process a new hole or cavity forms within a densely packed cellular region, creating what appears to be a new airspace. Brightfield time-lapse imaging of P4 lung PCLS showed formation of a solitary hole within a dense cellular region, the hole increased in diameter during the imaging period from 0 to 3.8 μm at 9 h and 8.5 μm at 19 h (Fig. 4a, red arrow and arrowheads, Supplementary movie 7, red arrowheads). Hollowing can also be seen by EpCAM-FITC/SiR-DNA labelled PCLS. Two cellular areas with only small holes visible at $t = 0$ gradually

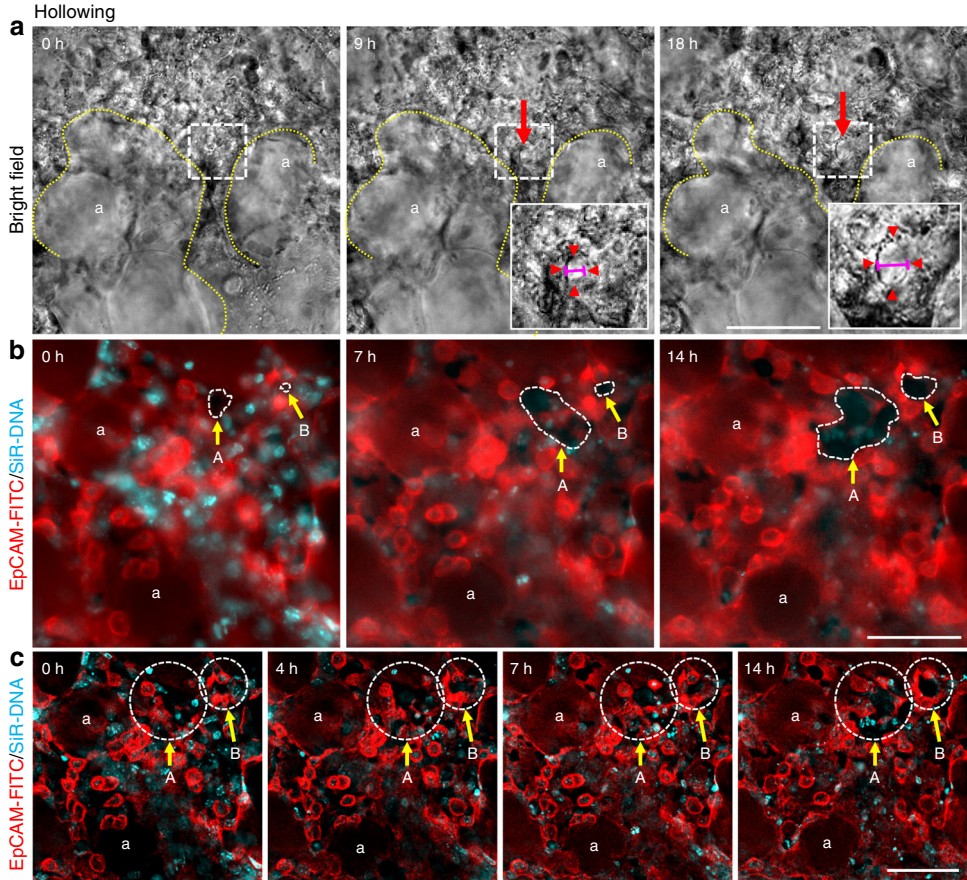

**Fig. 4** Cell hollowing in PCLSi. **a** Brightfield images from 18 h time-lapse video of P3 PCLS (Supplementary movie 7) shows 'hollowing' within the parenchyma at 0, 9 and 18 h (red arrow, and red arrowheads in white boxed area, dotted yellow lines outline surrounding airspaces). Inserts in **a** enlarged views of the white boxed area. Pink horizontal lines in insert depict width of 'hole' expanding (compare **a**, middle panel to **a**, right panel). Scale bar = 50 μm. **b** EpCAM-FITC (red)/SiR-DNA (cyan) labelled raw images of P3 PCLS from 14 h time-lapse video (Supplementary movie 8) show 'hollowing' at 0, 7 and 14 h (white dashed areas, A and B, yellow arrows). Scale bar = 50 μm. **c** Deconvolved still images from 14 h time-lapse video (Supplementary movie 9) show EpCAM +ve epithelial cell migration and rearrangement around the newly formed hollows/airspaces (white circles, A and B yellow arrows,). White 'a' indicates alveolar airspaces, n = 3 independent experiments using 3 separate mice, with duplicate slices from each mouse per experiment. Scale bar = 50 μm

resolve to form two separate, larger hollows, which are easily distinguished in raw time-lapse video and still images (Fig. 4b, areas A and B, Supplementary movie 8, white circles). Area A expands from a volume of 32.55 μm$^2$ at 0 h to 778.98 μm$^2$ at 14 h and area B from 2.55 μm$^2$ at 0 h to 160.76 μm$^2$ at 14 h. Deconvolution of this time-lapse sequence, revealed the hollow areas were encircled by EpCAM positive epithelial cells. (Fig. 4c, areas A and B, Supplementary movie 9, white circles). We speculate that the hollowing in video 7 is a new airspace being generated from a region previously occupied by cells and the hollowing seen in videos 8 and 9 are nascent alveoli that mature into deeper hollows generating wider airspaces as the movie proceeds.

To determine the frequency of septation, cell clustering and hollowing events seen in the movies at P3, we calculated the number of each type of event observed in a total of 12 separate PCLS fields from three different imaging experiments. The mean alveolar number per field of view was 5.66, and we observed the following frequencies of cell behaviours per field of view: septation, 1.9; clustering 5.5 and hollowing 1.9.

**Real-time imaging of PCLS reveals dynamic cell behaviour.** Analysis of brightfield time-lapse movies of PCLS revealed the rapid movement of non-structural cells, presumably neutrophils (Supplementary movie 10). In addition, we identified a particular cell behaviour that contributed to alveolar development, which we termed cell extension. This involves a cell extending around an existing alveolar wall (Fig. 5a–g, yellow arrow in a, b, d, red arrow in b and white dashed arrow in e–g, Supplementary movie 10). In this example, the cell extended an average of 1.29 μm/h over 16 h (Fig. 5c). Cell extension was less frequently observed than other cell behaviours; a total of 4 events in 12 PCLS fields. However, by analysing fluorescently labelled PCLS, we were also able to show the extension of an EpCAM positive cell, indicating that these rapidly extending cells are epithelial cells (Supplementary movie 11Cii, white arrow).

**PCLSi enables capillary network tracking in alveologenesis.** In addition to the epithelium, other cells are important in alveologenesis including fibroblasts and endothelial cells, which form the capillary network. Fluorescent labelling of PCLS with EpCAM-FITC and PECAM (Platelet Endothelial Cell Adhesion Molecule)-Alexa 647 enabled visualisation of both the epithelium and the capillary network at the same time (Supplementary movie 12). This dual labelling reveals the close interplay between the capillary endothelium and the alveolar epithelium during

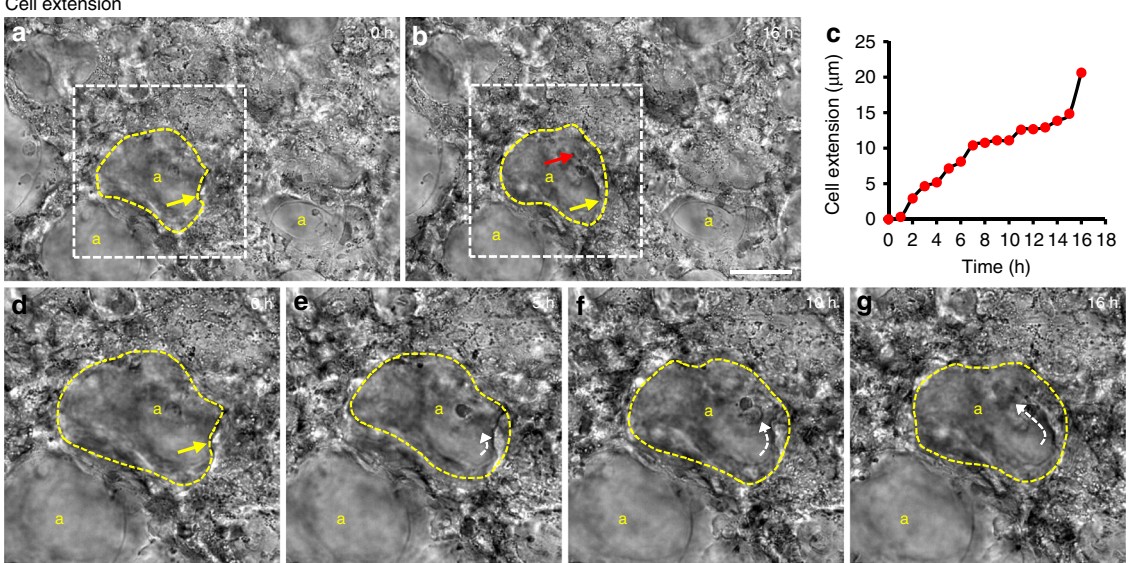

**Fig. 5** Cell extension during postnatal alveologenesis. Brightfield still images from P4 PCLS time-lapse video (Supplementary movie 10) at 0 (**a**, **d**), 5 (**e**), 10 (**f**) and 16 (**b**, **g**) hours showing cell extension along an alveolar wall (dashed yellow line in **a**, **b**, **d**–**g**); yellow 'a' indicates airspaces, yellow arrow indicates cell position at 0 h (**a**, **d**), red arrow indicates the final cell position (**b**). Measurement of cell extension during 16 h of time-lapse imaging (**c**). Enlargement of white boxed areas in **a** and **b** show the extending cell (white dashed arrow) during a 16 h video (**e**–**g**). Scale bar = 50 μm

alveologenesis. The close apposition of these cells can also be seen during septation where both cell types are visible in an extending septum (Supplementary movie 12 and zoomed in Supplementary movie 13).

**Abrogation of cell migration inhibits postnatal alveologenesis.** Next, to investigate the importance of cell migration in postnatal alveologenesis, we used para-Nitroblebbistatin (blebbistatin), a photostable derivative of blebbistatin that inhibits non-muscle myosin II[34]. We first optimised the dose of blebbistatin for these experiments (50 μM) and confirmed this was not toxic to PCLS (Supplementary Figure 6a, b). Blebbistatin (50 μM) was administered to P3 PCLS, pre-labelled with EpCAM-FITC, SiR-DNA and time-lapse imaging was conducted. Control treated PCLS (media with DMSO alone) showed normal epithelial cell migration as seen before (Fig. 6a–g, Supplementary movie 14A). However, blebbistatin markedly restricted EpCAM positive epithelial cell migration (Fig. 6h, i, Supplementary movie 14B). Interestingly, epithelial cells in blebbistatin-treated PCLS were not completely static, but, rather than moving from one place to another, cells appeared to agitate in situ (Supplementary movie 14B). Blebbistatin treatment also had a profound effect on EpCAM-FITC localisation; instead of a continuous band of staining around the edge of each cell, focal aggregations of fluorophore were present around the epithelial cell membranes (Fig. 6h, i). This feature was not due to blebbistatin toxicity, which was confirmed by MTT assay (Fig. 6j) but is probably a consequence of disruption caused by blebbistatin[35].

Quantitative analysis showed that blebbistatin treatment significantly inhibited mean net epithelial cell migration after 14 h compared with DMSO control treated samples (1.32 μm vs. 2.64 μm, respectively; $p < 0.0001$; Fig. 6k–m). Furthermore, 4% of highly motile epithelial cells migrated between 6 and 13 μm in 14 h in control, whereas, in blebbistatin samples, only 0.35% cells migrated between 6 and 7.5 μm (Fig. 6n; $p = 0.001$).

In addition to time-lapse imaging, we confirmed that blebbistatin (50 μM) treatment could inhibit alveologenesis in PCLS cultured for 72 h. Alveolar airspace number was quantified

by morphometric analysis as before (Fig. 2). In normal culture (SF-DMEM with DMSO), Lm decreased (93.03 μm vs. 61.38 μm at 0 and 72 h, respectively; $p < 0.001$; Fig. 7a, b) and the number of airspaces increased at 72 h (178 airspaces/mm[2] vs. 244 airspaces/mm[2] at 0 and 72 h, respectively; $p < 0.001$; Fig. 7a, c) reflecting active ex vivo alveologenesis; as previously reported by Pieretti et al.[19]. In contrast, there were no significant changes in Lm or airspace count between 0 h and 72 h in blebbistatin-treated PCLS (Fig. 7a–c).

Dual immunostaining of PCLS in culture with Ki67 and Sp-C antibodies showed no detrimental effects of blebbistatin on the percentage of either proliferation or ATII cells compared to DMSO control treated PCLS after 72 h of culture (Fig. 7d, e). A reduction in the percentage of Ki67 positive cells was noted after 72 h of culture in both control and blebbistatin-treated PCLS (Fig. 7e), but proliferation was unaltered at 24 h (Fig. 7f, g). We speculate that extended 3-day culture of PCLS may have some negative effects on cell proliferation, although cell survival (Supplementary Figure 7) and the percentage of ATII cells (21.03% vs. 20.21% in control and 19.05% vs. 21.27% in blebbistatin treatment, at 0 and 72 h, respectively) were not significantly affected (Fig. 7e). Crucially, despite the reduction in cell proliferation at 72 h, ex vivo alveologenesis still took place as previously seen[14]. Taken together, these data suggest that abrogation of cell migration by blebbistatin treatment inhibits murine alveologenesis in vitro.

Finally, we replicated a key set of experiments using a second modifier of the acto-myosin machinery, cytochalasin D (cyto-D), which inhibits actin polymerisation[36]. In DMSO treated control P3 PCLS, cell migration occurred as normal during 14 h of time-lapse culture (Fig. 8a, top panels, Supplementary movie 15A) whereas cyto-D treatment (100 ng/ml) led to marked inhibition of epithelial cell migration (Fig. 8a, bottom panels, Supplementary movie 15B). Cell tracking (Fig. 8b, c) and migration quantification (Fig. 8d, e) showed a significant reduction of net epithelial cell migration in cyto-D treated samples, when compared with DMSO control (1.44 μm vs. 2.32 μm, respectively; $p = 0.0022$; Fig. 8d). Furthermore, in control treated PCLS, 4.32% epithelial cells migrated between 6 and 14.3 μm, whereas none of the cells

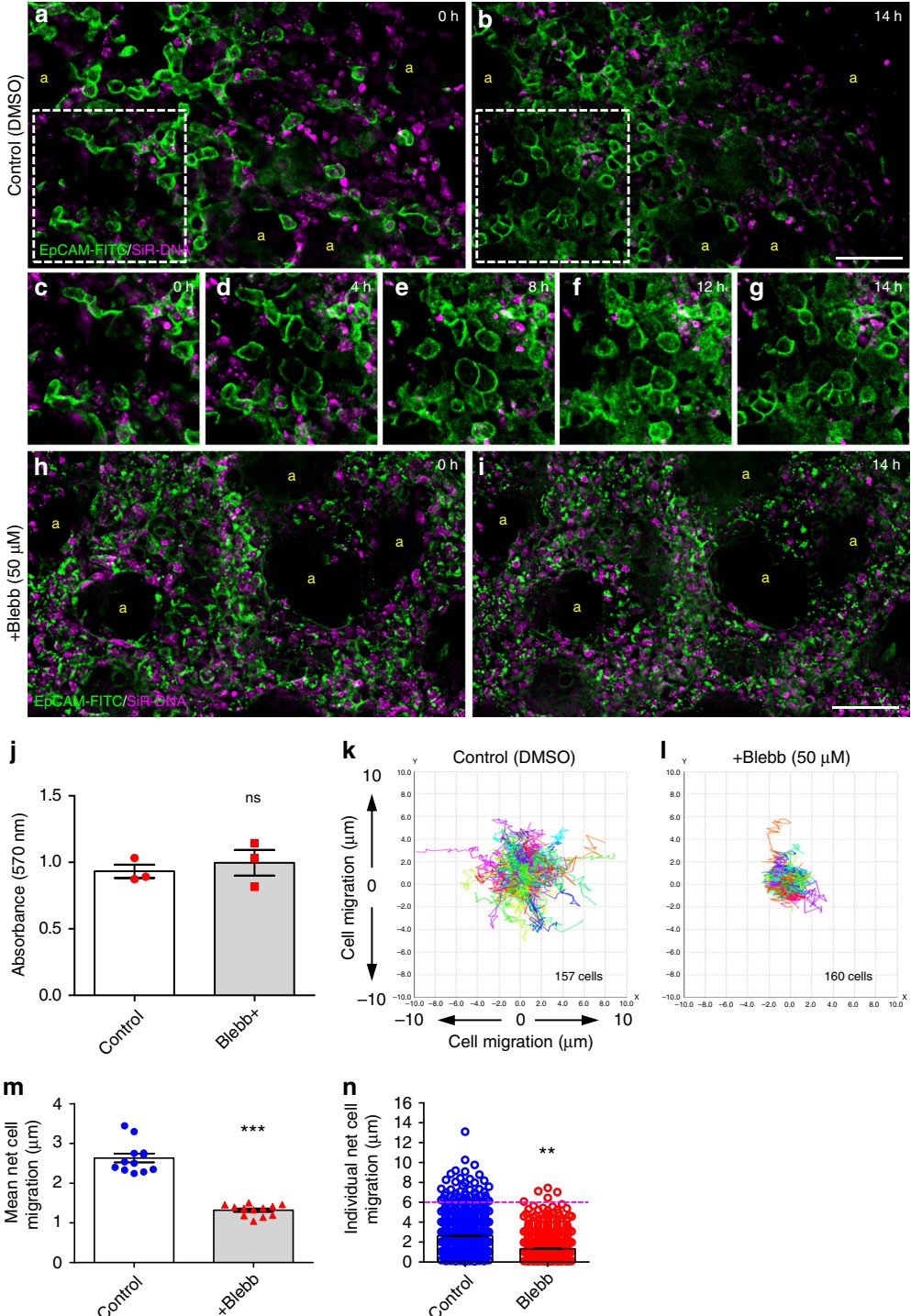

**Fig. 6** Blebbistatin inhibits epithelial cell migration in PCLS. Deconvolved widefield, single plane z-stack images from 14 h time-lapse videos (Supplementary movies 14A, B) of control (**a–g**) and blebbistatin-treated (**h**, **i**) P3 PCLS labelled with EpCAM-FITC (green) and SiR-DNA (magenta). Still images are taken from supplementary video 14A at 0 (**c**), 4 (**d**), 8 (**e**) 12 (**f**) and 14 (**g**) hours represent the white boxed regions in **a** and **b**. Scale bar = 50 μm. MTT cell viability assay comparing control and blebbistatin-treated P3 PCLS at the end of time-lapse, (**j**), n = 3 independent experiments using three separate mice, with duplicate slices per condition, per experiment, ns = not significant; paired Student t-test. Individual cell tracking over 14 h in a single field from P3 PCLS treated with control DMSO (**k**) or blebbistatin (**l**) containing media. Mean net epithelial cell migration in P3 control and blebbistatin-treated PCLS movies over 14 h (**m**). n = 3 independent experiments using three separate mice, with duplicate slices per condition, per experiment. Two fields were quantified per slice. Each dot represents mean net epithelial cell migration per field. Individual net cell migration in blebbistatin treated vs. DMSO control P3 PCLS after 14 h of time-lapse imaging (each dot represents a single cell) (**n**). A total of 1243 cells for DMSO control and 1340 cells for blebbistatin-treated PCLS were tracked. n = 3 independent experiments using three separate mice, with duplicate slices per condition, per experiment. Two fields were quantified per slice. ***p < 0.0001, **p = 0.001; Mann-Whitney U-test,. Yellow 'a' indicates alveolar airspaces. Error bars are defined as s.e.m

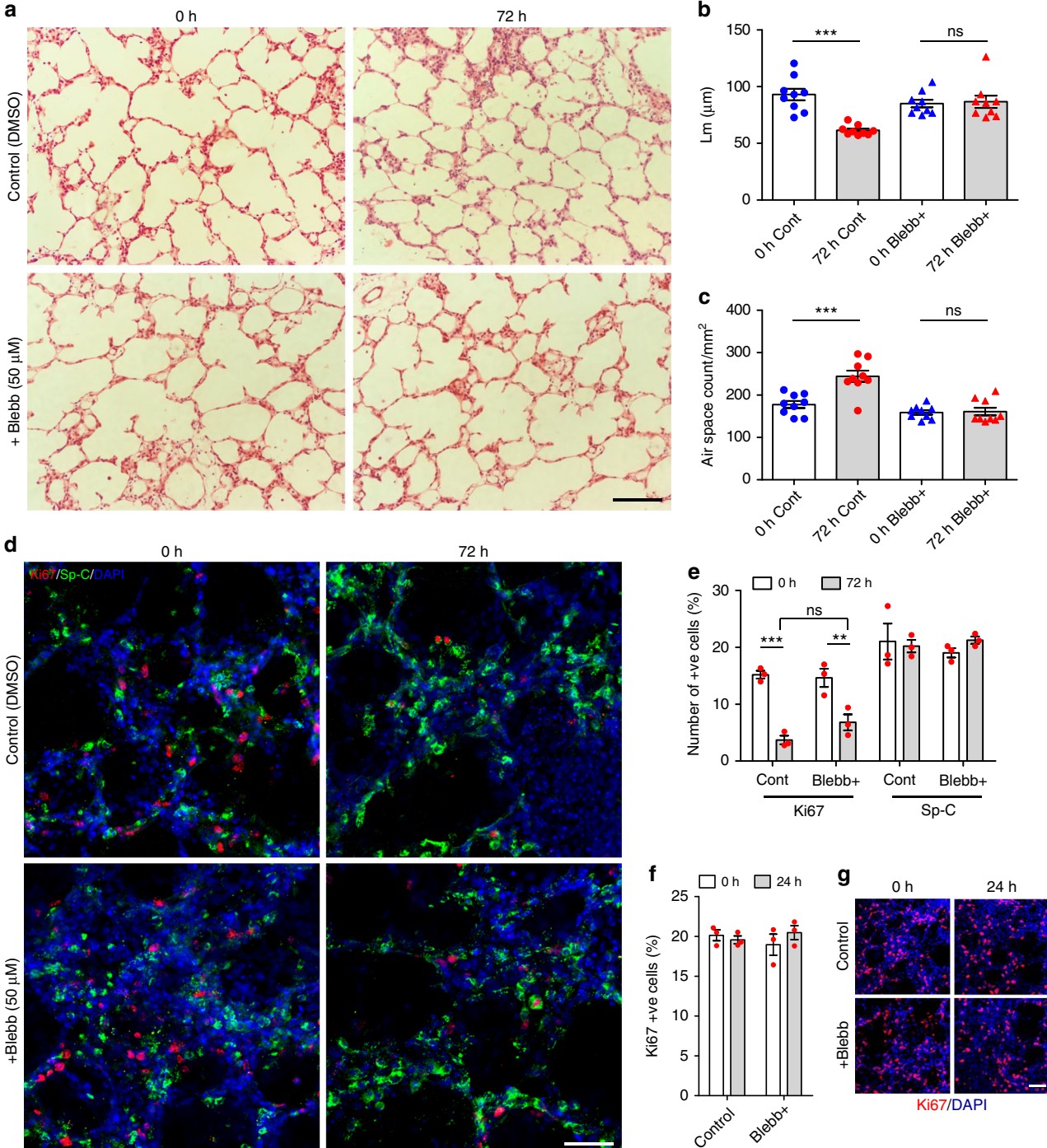

**Fig. 7** Blebbistatin inhibits ex vivo alveologenesis. H&E stained sections from P3 PCLS at 0 and 72 h, cultured with DMSO control (top panels) or 50 μM blebbistatin (bottom panels) (**a**). Scale bar = 100 μm. Mean linear intercept (Lm) (**b**) and airspace count (**c**) obtained from H&E sections of P3 PCLS treated with DMSO control or 50 μM blebbistatin at 0 and 72 h, n = 3 independent experiments using 3 separate mice, 3 H&E sections from each PCLS from each mouse were quantified per group, per experiment, each dot represents per field count (**b, c**); ***$p < 0.001$, ns = not significant, one-way ANOVA with Tukey's post hoc test. Confocal single plane z-stack images of DMSO control (top panels) and 50 μM blebbistatin (bottom panels) P3 PCLS at 0 and 72 h culture, immunostained with Ki67 (red), Sp-C (green) and DAPI (blue) (**d**). Scale bar = 50 μm. Quantification of Ki67 and Sp-C +ve cells in control and blebbistatin-treated P3 PCLS at 0 and 72 h culture (**e**), n = 3 independent experiments using 3 separate mice, with duplicate slices per group per experiment. Two fields were quantified per slice. Each dot represents mean value of per field counts per experiment; **$p < 0.01$; ***$p < 0,001$; ns = not significant; one-way ANOVA with Tukey's post hoc test. Quantification of Ki67 +ve cells in DMSO control and blebbistatin-treated P3 PCLS at 0 and 24 h in culture (**f**), n = 3 independent experiments using 3 separate mice, with duplicate slices per group, per experiment. Two fields were quantified per slice. Each dot represents mean value of per field counts, per experiment; one-way ANOVA with Tukey's post hoc test. Confocal single plane z-stack images of DMSO control (top panels) and 50 μM blebbistatin-treated (bottom panels) P3 PCLS at 0 and 24 h culture, immunostained with Ki67 (red) and DAPI (blue) (**g**). Scale bar = 50 μm. Error bars are defined as s.e.m

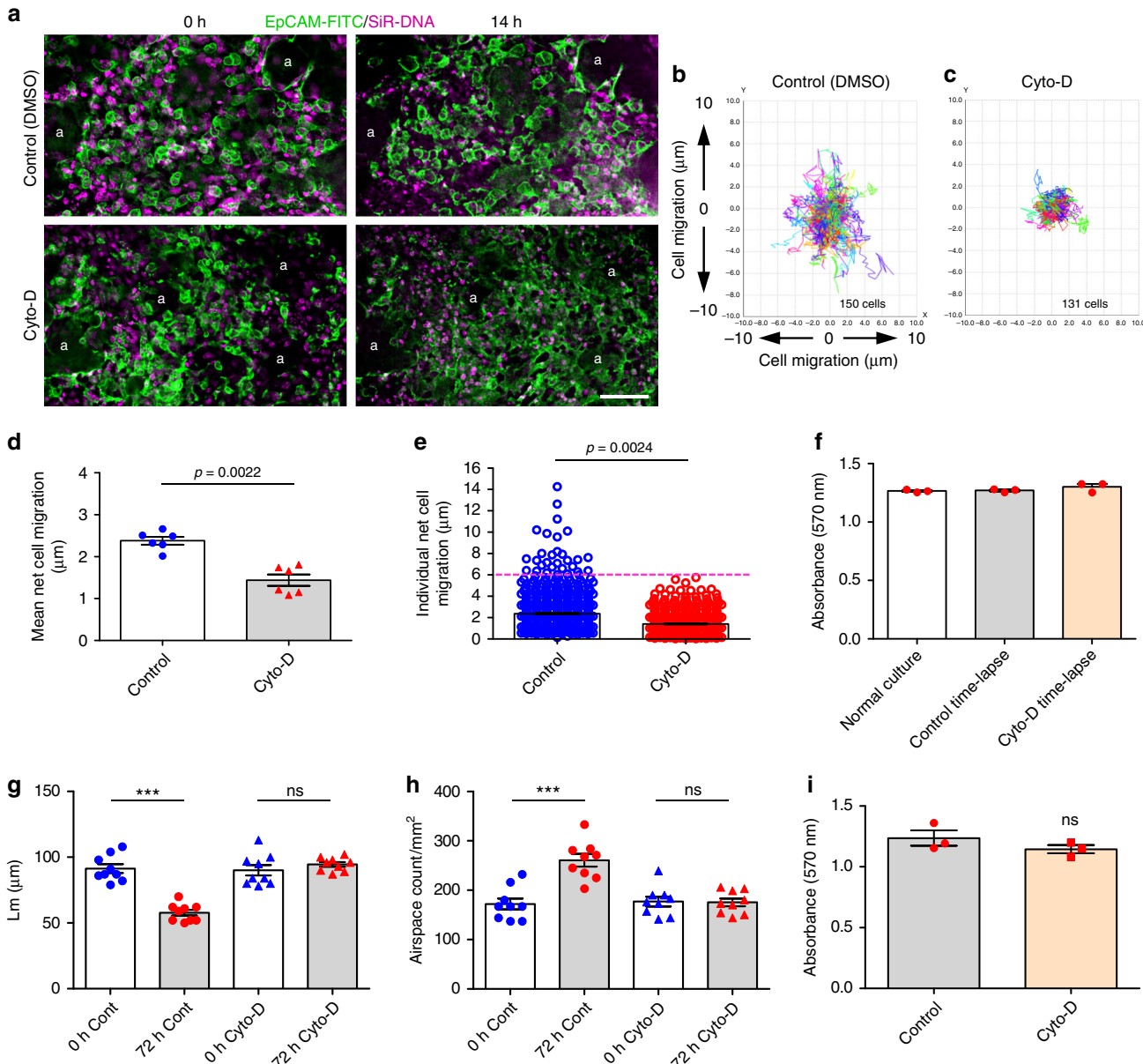

**Fig. 8** Cytochalasin D inhibits ex vivo alveologenesis. Deconvolved, widefield single plane z-stack images from 14 h time-lapse videos (Supplementary movies 15A, B) of control (**a**, top panels) and cyto-D treated (**a**, bottom panels) P3 PCLS labelled with EpCAM-FITC (green) and SiR-DNA (magenta) at 0 and 14 h. Individual cell tracking over 14 h in a single field from P3 PCLS treated with DMSO, control (**b**) or 100 ng/ml cyto-D containing media (**c**). Mean net epithelial cell migration over 14 h in DMSO control or 100 ng/ml cyto-D containing media (**d**), $n = 3$ independent experiments using three separate mice; two fields were quantified from each PCLS from each mouse per condition, per experiment. Each dot represents mean net epithelial cell migration per field. Mann-Whitney $U$-test, $p = 0.0022$. Individual net cell migration in cyto-D treated vs. DMSO control P3 PCLS after 14 h of time-lapse imaging (each dot represents a single cell) (**e**). A total of 778 cells in DMSO control and 718 cells in cyto-D treated P3 PCLS were tracked. Student $t$-test; $n = 3$ independent experiments using three separate mice, 2 fields were quantified from each PCLS from each mouse per condition, per experiment, $p = 0.0024$. MTT cell viability assay comparing P3 PCLS after 15 h under normal culture conditions or after 15 h time-lapse imaging in DMSO control or cyto-D containing media (**f**), $n = 3$ independent experiments using 3 separate mice, quantification was from a single PCLS per condition, per experiment, one-way ANOVA with Tukey's post hoc test. Mean linear intercept (Lm) (**g**) and airspace count (**h**) from H&E sections of P3 PCLS at 0 and 72 h of ex vivo culture; $n = 3$ independent experiments using 3 separate mice; 3 H&E sections from each PCLS from each mouse were quantified per group, per experiment, each dot represents per field count (**g**, **h**); ns = not significant, ***$p < 0.001$; one-way ANOVA with Tukey's post hoc test. MTT cell viability assay on P3 PCLS at 0 and 72 h of culture in DMSO control or cyto-D containing media (**i**), $n = 3$ independent experiments using three separate mice, quantification was from a single PCLS per condition, per experiment; ns = not significant; paired Students $t$-test. Error bars are defined as s.e.m

migrated ≥6 µm in cyto-D-treated PCLS ($p = 0.0024$, Fig. 8e). Cyto-D treatment did not cause significant cell toxicity (Fig. 8f).

In line with blebbistatin data, morphometric analysis (Lm and airspace count) showed that cyto-D treatment of PCLS for 72 h significantly restricted in vitro alveologenesis, as shown by

unaltered Lm (90.11 µm at 0 vs. 94.56 µm at 72 h; Fig. 8g) and airspace count (177 airspaces/mm² at 0 vs. 175.4 airspaces/mm² at 72 h; Fig. 8h) in P3 PCLS after 72 h culture; compared to DMSO control PCLS where Lm was significantly reduced ($p < 0.001$; Fig. 8g) and airspace count increased by 1.5-fold in 72 h

compared to $t = 0$ ($p < 0.001$; Fig. 8h) as expected. No significant cell toxicity was observed after 72 h cyto-D treatment (Fig. 8i).

## Discussion

We have established a method for visualising murine alveologenesis in real time. Alveologenesis occurs over a relatively long time period; to capture this process, we conducted long-term movie imaging of PCLS (PCLSi) in which cells were labelled with a combination of live cell dyes and conjugated antibodies. We confirmed that despite the likelihood that at least some agarose is retained in the airspaces of PCLS[23], alveologenesis is able to occur, in agreement with a previous study of alveologenesis in static PCLS[19]. The presence of agarose in PCLS, prevents the alveoli from collapsing and in this way mimics the in vivo alveolar environment. To enable individual cell dynamics to be distinguished throughout the videos, we used deconvolution of time-series images to reduce out-of-focus light and enable single cell tracking. Imaging thick post-natal tissue samples for such a long time period distinguishes the PCLSi model from other live-imaging techniques. Importantly, the combination of cell labelling with widefield microscopy and deconvolution avoids the need for expensive confocal microscopes whilst also providing a widely accessible alternative to using transgenic reporter mouse lines or complex in vivo imaging techniques.

In the current study, PCLS were maintained in a heated and humidified chamber with room air oxygen levels during imaging (37 °C, 5% $CO_2$, 20–21% oxygen). As the PCLS are submerged in media, oxygen tension is expected to be lower than in air but we did not measure this parameter in the media. In future, it would be interesting to investigate the effect of altered oxygen levels on alveologenesis using this technique, given the differential effects demonstrated in other models[37–39].

Elucidating the mechanisms of alveolar formation is essential not only to understand how dysregulation of this process leads to disease but also to develop novel regenerative treatments for damaged alveolar tissue. Data from previous studies, derived from static samples, has identified septation as the predominant event in alveologenesis. Secondary septation events were observed in this study and fluorescent labelling of specific cell populations enabled us to visualise the dynamics of alveolar epithelial cells with the capillary endothelium during septation, in real-time. Imaging of lung slices also revealed that a proportion of epithelial cells were highly motile during alveologenesis.

We identified several cell behaviours that contribute to alveologenesis. Based on careful analysis of all our movies, we propose that these behaviours represent different stages or strategies for alveolar development from de novo formation to terminal maturation. Cell clustering, where epithelial cells migrate to form a tight cluster of cells in one area, appears to represent an early stage in alveolar development (Fig. 9a) and this was the most frequent cell behaviour seen in our movies. Hollowing initially occurs within a cell cluster; at the beginning of this process the cluster of epithelial cells rearranges to enable a new hole to open up (Fig. 9b). Subsequently, further hollowing occurs to establish a 3-dimensional pocket, this includes broadening of the initial small hole to form an airspace (Fig. 9b). Once an alveolus is formed, further epithelial cells can migrate towards this and integrate into the nascent alveolus contributing to the final structure (Fig. 9b). We frequently observed epithelial cells migrating towards existing airspaces and contacting another cell already around the airspace.

The final cell behaviour we discovered was cell extension, where a single cell elongates around an existing airspace (Fig. 9c). We only observed cell extension where a morphologically mature alveolus was present; we, therefore, speculate that cell extension only occurs at a late stage of alveologenesis, when an intact alveolar wall has formed. Our identification of an EpCAM positive cell extending round an alveolar wall raises the interesting possibility that these elongating cells could be epithelial cells differentiating into type I alveolar cells but further studies will be required to test this hypothesis.

We found that treatment of PCLS with blebbistatin and cytochalasin D, both of which interfere with cell migration, severely

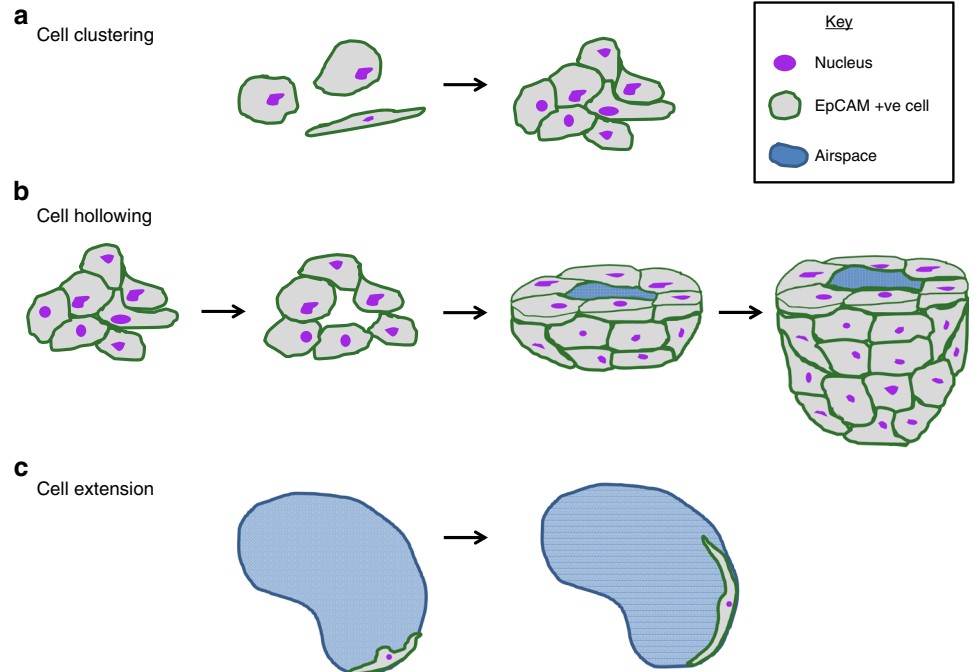

**Fig. 9** Diagram depicting cell behaviours identified in time-lapse videos. Schematic to show migratory epithelial behaviours that contribute to different stages of alveolar development. **a** Epithelial cells migrate and form clusters; **b** a small hole forms within a cell cluster, this widens and deepens to form a hollow following which additional epithelial cells migrate and integrate into the alveolus (g); **c** a cell extends around the wall of a mature alveolus

restrict alveologenesis. Blebbistatin selectively inhibits non-muscle myosin II and therefore affects cell contraction, which is needed to drive cell migration. Cytochalasin D inhibits actin polymerisation and migration associated events like accumulation of actin at the leading edge of cells. At the dose used, blebbistatin did not affect cell proliferation, therefore, we were able to assess the relative contributions of cell migration and proliferation to alveologenesis in blebbistatin-treated PCLS. Of note, our data revealed that the percentage of epithelial cell proliferation in ex vivo PCLSi, is relatively low during bulk alveologenesis, compared to previous studies in vivo[22] and our results indicate that cell migration and not cell proliferation is the dominant driver of alveologenesis. Firstly, at the doses of blebbistatin used in our PCLSi cultures, cell movement was almost completely inhibited but there was no change in the percentage of proliferation within PCLS at 24 h. Secondly, we found that alveologenesis was inhibited in PCLS cultured for 72 h with blebbistatin but continued in control treated cultures. Despite the difference in alveologenesis between control and inhibitor-treated PCLS, we found a marked decrease in proliferation at 72 h in both these groups, indicating that it is not proliferation that affects alveologenesis.

Our data describing intervention with cytoskeleton inhibitors shows that this model can be used to manipulate the alveolar environment and assess the resulting impact on alveologenesis. In this manuscript, we have focused on investigating the role of epithelial cells in alveologenesis and we have also shown that the capillary endothelium can be tracked. In future this technique could be used for more detailed investigation of the capillary endothelium and to investigate the role of other cell populations such as fibroblasts in alveologenesis. Moreover, the PCLSi technique could be used with tissue from genetically modified mice to investigate genetic and cellular contributions to alveologenesis. This technique could also be adapted for tissue slices from other organs and species, including humans.

We have identified dynamic epithelial cell behaviours that contribute to murine alveologenesis and we show that cell migration is the dominant driver of this process. Combining postnatal tissue slices with time-lapse imaging provides a powerful tool for future research in the lung and other organs.

## Methods

**Mice**. All animal maintenance and procedures were carried out according to the requirements of the Animal (Scientific Procedures) Act 1986. Animal work was approved by the South Kensington and St. Mary's AWERB committee, Imperial College London.

**Precision-cut lung slicing and in vitro culture**. Precision-cut lung slices (PCLS) were obtained from mouse lungs according to the protocol described in ref. [19], with some modifications. Post-natal day 3 (P3), P7, P14 and adult C57BL/6 mice were humanely killed by intraperitoneal injection of pentobarbital. The anterior chest wall was excised and trachea was carefully exposed. A tiny opening was made in the anterior wall of the trachea just below the cricoid cartilage. A rigid metallic cannula (25G for P3, 23G for P7 and 21G for P14 and adult mice) was carefully inserted through the trachea up to a millimetre above the bifurcation of the principal bronchi and fixed in place by suture. After cannulation, the lungs were inflated with 37 °C 1.5% low-melting-point agarose (Sigma; Cat. No. A9414) prepared with 1× HBSS/HEPES buffer (Life Technology; Cat. No. 14025-050). Agarose was injected to inflate both lungs keeping them in situ within the chest cavity at volume that enabled lungs to be fully inflated without hyper- or sub-optimal inflation (P3, 0.2 ml, P7, 0.275 ml, P14 0.35 ml and adult 1 ml agarose for PCLS or PBS for histology). These volumes were used for respective age groups throughout all experiments. After inflation, agarose was solidified by applying ice to the chest cavity for 1 min. Subsequently, the lungs were excised from the body along with heart and trachea and immersed in ice-cold serum-free DMEM (SF-DMEM) (Life Technologies; Cat. No. 31966-021), and kept on ice until slicing. Left lung lobes were isolated and cut transversely at 300 µm using an automated vibratome (Compresstome® VF-300-0Z; Precisionary Instruments LLC) in ice-cold HBSS/HEPES buffer. Slices were obtained from the middle 2/3rds of the lobe, to ensure similar sized slices, and placed in a 24-well plate in ice-cold SF-DMEM for all experiments. Using this technique, a P3 left lung provides 12 slices and an adult lung provides approximately 36 slices. PCLS were

then incubated at 37 °C for 2 h and washed twice with warm SF-DMEM to remove excess agarose from the tissue. Slices were incubated for a further 1 h in SF-DMEM at 37 °C. To select the optimal media for PCLS culture, adult PCLS were cultured for up to 96 h in presence of 5% $CO_2$ and 95% air at 37 °C in either SF-DMEM or in M199 media (Life Technologies; Cat. No. 11150059) supplemented with Insulin/transferrin/selenium (Life Technologies; Cat. No. 41400045), vitamin C (Cat. No. A4403), vitamin A (Cat. No. R2625), hydrocortisone (Cat. No. H4001) (All from Sigma) as described previously ref. [19]. Both media were supplemented with 1% penicillin–streptomycin (Life Technologies; Cat. No. 15140122) and media was changed every other day.

**Viability assays**. To assess cell viability and metabolic activity, lung slices were collected following 1 h, 24 h, 48 h, 72 and 96 h in culture. Cell viability was assessed by MTT, Caspase, and Live/Dead assays.

MTT assay: To assess metabolic activity within cells, for each experiment, similar size PCLS were placed into wells of a 24-well plate in duplicate (1 slice per well). MTT assay was performed according to manufacturer's instructions (Sigma; Cat. No. M2128). Briefly, 10% MTT solution (Stock Con. 5 mg/ml) was prepared in SF-DMEM and 500 µl solution was added per well, per slice and incubated at 37 °C for 1 h. Subsequently, formazan crystals that formed within the viable cells were solubilised by adding an equal volume of DMSO for 10 min and incubating at 37 °C for 10 min. 200 µl of eluted formazan solution from each slice was placed into individual wells of a 96-well plate. Absorbance (OD) was measured at 570 nm and corrected at 690 nm using a plate reader.

Caspase assay: To detect caspase-3 activity, marking apoptotic cells in PCLS, NucView 488 Caspase-3 assay kit was used following manufacturer's instructions (Biotium; Cat. No. 30029-T). Briefly, PCLS were plated in 24-well plate (1 slice/well) and 250 µl of 5 µM NucView 488 substrate prepared in SF-DMEM media was added and incubated in the dark at room temperature (RT) for 30 min. Slices were then washed 3 times with HBSS and fixed with 10% buffered formalin for 30 min in the dark, then washed 3 times and counterstained with DAPI (Sigma; Cat. No. D9542) at 1:500 dilution (Stock concentration 10 mg/ml) for 30 min. Slices were mounted on glass slides and images were captured with a Zeiss Axio Observer widefield microscope using a ×20, 0.8 NA air objective and Zen2 acquisition software, blue version.

Live/Dead assay: Cell viability in PCLS was assessed using LIVE/DEAD® Viability/Cytotoxicity Kit following manufacturer's instruction (ThermoFisher Scientific; Cat. No. L3224). PCLS were incubated with 2 µM Calcein AM and 2 µM Ethidium homodimer-1 (EthD-1) in 250 µm HBSS for 30 min at 37 °C. Then washed two times with HBSS and fixed with 10% buffered formalin for 30 min at RT and washed. PCLS treated with 70% methanol for 30 min at RT were used as a positive control for dead cells and media treated PCLS were used as a positive control for live cells. Samples were mounted on glass slides with ProLong® Gold Antifade Mountant (ThermoFisher Scientific; Cat. No. P36930) and imaged using a Zeiss LSM-510 confocal microscope with a ×20 0.8 NA air objective and ZEN 2009 (black edition) software.

**Live cell staining of PCLS and time-lapse imaging**. For dual live cell staining with EpCAM-FITC, SiR-DNA, PCLS were incubated for 1 h at 37 °C with FITC-conjugated EpCAM antibody at 1:200 (eBioscience; Cat. No. 11-5791-80; Clone G8.8) and silicon rhodamine far-red fluorophore-conjugated DNA minor groove binder bisbenzimide (SiR-DNA) at 1:300 (tebu-bio ltd; Cat. No. SC007) or Alexa-647 conjugated PECAM antibody (CD31-Alexa 647; Biolegend; Cat No 102416; Clone 390) at 1:200 with EpCAM-FITC in 500 µl SF-DMEM per sample (Life Technologies; Cat. No. 21063029). Slices were then washed three times with SF-DMEM and for imaging, were either immediately imaged or fixed with 10% buffered formalin and stored at 4 °C for later imaging. For time-lapse imaging, the EpCAM-FITC, SiR-DNA or EpCAM-FITC, PECAM labelled PCLS were immediately placed at the centre of the well of an uncoated ibidi 24-well µ-plate (ibidi; Cat. No. 82401). Then a 0.4 µm pore, 12 mm transwell (Corning; Cat. No. 3460) was placed onto the PCLS, with the rim of the transwell removed to allow the transwell filter to contact the slice at the bottom of the well (Fig. 1a). Image media, containing EpCAM-FITC at 1:500 and SiR-DNA at 1:1000 or PECAM at 1:500 in phenol-free SF-DMEM, was then added to the upper chamber (500 µl) and bottom chamber (300 µl). Finally, to keep the PCLS in place, a 1.66 g metal flat washer (M8- 5/16th inches diameter) was placed on top of the transwell housing (Fig. 1a). The PCLS were then left in the incubator for 2 h. This incubation step ensures the cells are all labelled and allows the slice to settle down prior to image acquisition. The 24-well plate was then transferred to a pre-equilibrated and humidified incubator chamber of an inverted Zeiss Axio Observer widefield epifluorescence microscope and PCLS were kept in the following conditions: 37 °C, 5% $CO_2$ and room air oxygen levels, approx. 21%. Imaging was conducted using a long working distance ×40 (0.7 NA, air) objective lens. The 2 h incubation and pre-equilibration of the microscope incubator were crucially important to avoid the plane of focus drifting during time-lapse image acquisition. Slices that were subject to brightfield imaging only, were not stained but were otherwise prepared and set up as for stained slices. In each experiment, a minimum of duplicate slices for each experimental group, were imaged. A maximum of four slices were imaged in a single time-lapse experiment e.g., two in DMSO control media and two blebbistatin or cyto-D treated. Images were captured using brightfield, GFP filter, excitation

450–490 nm, emission 500–550 nm (for EpCAM-FITC) and Cy-5, excitation 625–655 nm, emission 665–715 nm (for SiR-DNA) from 4 fields per slice for 8–19 h at 15 min intervals or 64 h at 60 min intervals. Eleven images were captured along the z-axis with 1 μm step-gap to make a z-stack from each slice.

**Staining of PCLS.** PCLS were dual stained with EpCAM-FITC, SiR-DNA as above and fixed with 10% buffered formalin. To prevent non-specific binding and to permeabilise the tissue, fixed PCLS were then incubated in PBSBT (1% BSA; Sigma, Cat. No. A7030 and 0.5% Triton-X 100; Sigma Cat. No. X100 in PBS) for 1 h at RT in the dark. Samples were then stained with anti-mouse CD11c-PE primary antibody (Biolegend; Cat. No. 117307; Clone N418) at 1:200 to label macrophages or with rabbit anti-pro-surfactant protein-C (Sp-C) antibody (Millipore; Cat. No. AB3786) or anti-pan-cytokeratin antibody (Sigma; Cat. No. C2931; Clone C-11) at 1:250 in PBSBT and incubated at 4 °C overnight. After incubation, EpCAM-FITC, CD11c-PE, SiR-DNA labelled samples were washed and mounted onto glass slides with ProLong® Gold Antifade reagent. For Sp-C and pan-cytokeratin samples were washed and incubated with goat anti-rabbit IgG secondary antibody, Alexa Fluor 568 (Invitrogen; Cat. No. A-11011) and goat anti-mouse IgG Alexa 647, respectively at 1:250 dilution in PBSBT for 1 h at RT. EpCAM-FITC, Sp-C, SiR-DNA stained slices were then mounted onto glass slides as above. EpCAM-FITC, Pan-cytokeratin dual labelled slices were counter-stained with DAPI and mounted on glass slides as above. For Sp-C, Cd11c-PE dual labelling, goat anti-rabbit IgG, Alexa Fluor 488 (Invitrogen; Cat. No. A-11008) antibody was used to visualise Sp-C at 1:250 and counterstained with DAPI and mounted as above. For DAPI and Ki67 or Sp-C staining, PCLS were first fixed with 10% formalin, blocked and permeabilised as above. Samples were then incubated with a combination of anti-mouse and rat Ki67 primary antibody (eBioscience; Cat. No. 14-5698-82; Clone SolA15) at 1:500 and Sp-C antibody at 1:250 at 4 °C overnight. Primary antibodies were visualised using secondary antibodies goat anti-rat IgG, Alexa Fluor 568 (Cat. No. A-11077) for Ki67 and goat anti-rabbit IgG, Alexa Fluor 488 (Cat. No. A-11008) for SP-C, both from Invitrogen at 1:250 dilution. Nuclear staining was achieved by incubating slices with DAPI for 30 min and samples were then mounted onto glass slides as above. To confirm that there was no non-specific binding of conjugated fluorophores used for live-cell time-lapse imaging, unlabelled P3 mouse lung slices were stained with secondary isotype IgG antibodies: rat IgG-FITC (eBioscience; Cat. No. 11-4321-42) for EpCAM-FITC or Armenian hamster IgG-PE (Biolegend; Cat. No. 400907) for CD11c or rat IgG-Alexa Fluor 647 (Biolegend; Cat. No. 400526) for PECAM at 1:200 for 1 h at RT and counterstained with DAPI and then mounted on glass slides as above. Images were captured using either Zeiss Axio Observer inverted microscope, with Lumencor Spectra X LED light source and Hamamatsu Flash 4.0 camera, using ×40, 0.7 NA, air objective (for EpCAM-FITC, CD11c-PE, SiR-DNA and PECAM or Zeiss LSM-510 inverted confocal microscope, using ×20, 0.8 NA, air objective (for Ki67 and Sp-C; and for secondary isotype IgG antibodies).

**Blebbistatin and cytochalasin-D treatment of PCLS.** To investigate the effects of blebbistatin and cytochalasin-D on ex vivo alveologenesis, P3 PCLS were treated with 50 μM para-Nitroblebbistatin (blebbistatin, Cayman Chemical Company; 13891) or 100 ng/ml cytochalasin D (cyto-D, Sigma; C8273). To evaluate the inhibitors' effects on cell migration, 12–19 h time-lapse imaging experiments were conducted. For this, PCLS were first labelled with EpCAM-FITC and SiR-DNA live dyes and prepared for imaging as previously described. 50 μM blebbistatin or 100 ng/ml cyto-D in image media was added to the slices (500 μl in upper chamber, 300 μl in bottom chamber of the transwell). Slices were then incubated at 37 °C for 2 h prior to beginning time-lapse acquisition, as previously described. Blebbistatin and cyto-D were reconstituted using DMSO and the final concentration of DMSO in the working media was 0.2% and 0.01%, respectively. Therefore control media for blebbistatin and cyto-D was phenol-free SF-DMEM containing 0.2% or 0.01% DMSO, respectively. In some time-lapse experiments, blebbistatin and cyto-D were used on un-labelled PCLS for brightfield imaging only. To evaluate the effects of blebbistatin and cyto-D ex vivo alveologenesis, cell proliferation and Sp-C positive ATII epithelial cells, un-labelled P3 PCLS were cultured in SF-DMEM supplemented with 50 μM blebbistatin or 100 ng/ml cyto-D for 72 h under normal culture (non-imaging) conditions (37 °C, 5% $CO_2$). Media was changed every alternate day. DMSO controls were as described for time-lapse experiments. Slices were then fixed with 10% formalin as above for histology sectioning and morphometric analysis.

**Morphometric analysis of lung tissue.** To evaluate alveologenesis, mean linear intercept (Lm) and the number of airspaces were quantified in H&E stained lung sections from P3 PCLS cultured for 72 h or left lobes of lungs from different post-natal age groups following histological staining as follows: P3 PCLS at t = 0 and t = 72 h culture were fixed with 10% buffered formalin for 30 min, embedded in paraffin, keeping the slice surfaces horizontal to the block and sectioned at 4 μm thickness. Sections were deparaffinised and stained with Haematoxylin and Eosin. Three separate experiments were conducted with three individual mice.

For whole lung processing, lungs from P3, P7, P14 or adult mice were inflated with PBS using age-specific, defined volumes (described above) and fixed in 10% formalin overnight at 4 °C. Following fixation, left lung lobes were paraffin embedded, sectioned along the long axis and H&E stained, as above. Quantification

was carried out on three mice for each age group. Images were captured using a Leica DM2500 widefield microscope and a ×20, 0.7 NA objective. For quantification of Lm, a grid of eight horizontal lines was superimposed on images from H&E sections, using FIJI (ImageJ, version 2.0)[40]. The number of times alveolar cells intercepted the line was counted and Lm was calculated using the following equation: Lm = NL/X, where N = number of lines counted, L = length of line, and X = total number of intercepts counted. A minimum of three fields per lung section, and three sections per PCLS or per left lobe were imaged. In the graphs, each dot represents per field count. The number of alveolar spaces/airspace per field was counted and presented as number of airspaces per millimetre square area. Alveolar spaces were quantified from the same visual fields used for Lm quantification. Fields of view containing blood vessels or airways were omitted from analysis.

**Quantification of cell behaviours.** To quantify cellular events, such as, cell clustering, hollowing, septation and cell extension that observed during time-lapse imaging (×40 objective), the number of times each event was observed in a single field of view obtained from a movie file was recorded. A total of 12 time-lapse fields from three separate experiments were examined. The frequency of each type of cellular behaviour was determined by expressing the total number of observations of that behaviour/number of fields analysed. The mean number of alveoli per field of view was 5.66.

**Cell tracking and proliferation analysis using Icy software.** Icy (version 1.9.8.0), an open source bioimaging analysis software, created by the Quantitative Image Analysis Unit at Insitut Pasteur, Paris, France, was used for cell tracking and migration quantification during time-lapse imaging[33]. EpCAM-FITC, SiR-DNA labelled raw time-lapse sequences from each field of a lung slice were up-loaded into Icy and EpCAM-positive cells were tracked. Results were manually checked to ensure tracking was correct.

Cell tracking: EpCAM-FITC positive cells were tracked for quantification of epithelial cell migration[41]. FITC-channel was selected from best-focused slice from a z-stack from each slice. Cell migration in the X–Y plane only (not the Z-plane) was recorded. We focused on two aspects of tracking data: (1) Net cell migration within a specified time-lapse period, in which the mean value was calculated to present how much linear distance in the X–Y axis (Initial point A to end point B) a cell had migrated. A mean value of migrated distances of total number of cells from each field is presented as mean net cell migration, (2) we also looked at the net distance travelled by individual cells in a field within a specified time period in each experimental group. The same duration was used for all groups within one experiment enabling direct comparison to be made between groups within that experiment. Within the PCLS a majority of cells were relatively sessile and migrated less than 2 μm. However, a subpopulation of epithelial cells were highly motile. To determine the proportion of these highly motile cells we quantified the percentage of cells that migrated between 6 and 14 μm per field within a defined time period (8 h for Fig. 2 and 14 h for Figs. 6 and 8). Two to four fields from each slice were analysed. Each dot in the graphs represents either mean value of migration distances of total cells per field for (mean net cell migration graphs) or the distance a single cell migrated (individual net cell migration graphs) as specified in the figure legends. Three independent experiments were conducted using three mice per group/age. Using this strategy, cell migration was analysed in time-lapse sequences of 8 h in P3, P7, P14 and adult mice PCLS, and 14 h duration for blebbistatin/cyto-D treated and control groups as described in previous sections.

Cell proliferation: The Spot detection tool in Icy was utilised to quantify the percentage of Ki67 and Sp-C positive cells within fixed PCLS samples following 24 and 72 hr culture in presence and absence of blebbistatin, using confocal images of Ki67/Sp-C/DAPI immunostained slices. At least four fields from each PCLS at each time point were selected for quantification. Three experiments were conducted on PCLS prepared from three P3 mice.

**Manual quantification of cell proliferation in vivo and in PCLS.** To quantify the percentage of cell proliferation in the alveoli at different developmental ages, lungs from P3, P7, P14 and adult mice were inflated with PBS using age-specific, defined volumes (described above) and fixed in 10% formalin overnight at 4 °C. Following fixation, left lung lobes were paraffin embedded, sectioned along the long axis and immunostained with anti-mouse/rat Ki67 primary antibody (eBioscience; Cat. No. 14-5698-82; Clone SolA15) at 1:500 at 4 °C overnight. Secondary treatment was done using HRP-conjugated anti-rat IgG (VECTASTAIN Elite ABC HRP Kit; Cat. No. PK-6104) and visualised using DAB Substrate Kit (BD Pharmingen; Cat. No. 550880) following manufacturer's instructions. Nuclei were counterstained with haematoxylin. 200 cells were counted along the alveolar wall per section using a Leica DM2500 widefield microscope and a ×20, 0.7 NA objective. Percentage of Ki67 positive cells was calculated. Quantification was made on three sections from each lung, and three lungs from each age group (n = 3 mice each group).

For quantification of proliferation in PCLS, slices were stained with Ki67, EpCAM-FITC and DAPI as previously described. The total percentage of proliferating cells was determined by manual counting of the total number of Ki67 positive cells per field and expressing this as a percentage of total cells per field. The percentage of proliferating epithelial cells was calculated by counting the number

of dual EpCAM, Ki67 positive cells per field and expressing this as a percentage of the total number of Ki67 positive cells per field. Counting was done manually at ×40 magnification and 6 separate fields of view were counted at P3 and P7 with at least 200 cells per field.

**Image deconvolution and movie generation**. Z-stacks, either single; static or multiple; time-lapse, were acquired by widefield microscopy. Each z-stack consisted of 11 separate z-slices per sample, with a 1 μm step between slices. Static and time-lapse z-stack images were deconvolved using Huygens deconvolution software from Scientific Volume Imaging (SVI, Essential version 17.10). The automated CMLE algorithm was used for all image deconvolution in this study. Post-deconvolved, time-lapse z-stacks were then imported into Icy to generate videos (.avi file format). 2D videos and static images shown in the manuscript are from the best-focused single z-plane within a z-stack. For 3D video reconstruction, deconvolved z-stack time-lapse files were imported into NIS-Elements (Version 4.50, Nikon Instruments, UK) and aligned to correct for X–Y drift. 3D movies were generated using the 'Volume View' and 'Movie Maker' modules of NIS-Elements. Raw image files were used to generate brightfield time-lapse movies.

**Statistical analysis**. Data were analysed using GraphPad Prism version 5 or Excel (Office 2011 version). Data are presented as the mean values for each experimental group with variation represented as ± SEM (standard error of mean). Differences between two groups were tested using Mann-Whitney $U$-test or paired two-tailed Student's $t$-test. Differences between three or more groups were tested using One-way ANOVA with Tukey's post hoc test. A '$p$' value <0.05 was considered statistically significant. The number of experiments ($n$ number) is noted in the respective figure legends throughout the manuscript.

**Reporting summary**. Further information on experimental design is available in the Nature Research Reporting Summary linked to this article.

## Data availability

The data that support the findings of this study are available either in the manuscript or supplementary files. Any additional data from this study are available from the corresponding author upon reasonable request.

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

## Acknowledgements

This project was funded by a grant from the Leverhulme Trust to CHD (RPG-2015-226). The Facility for Imaging by Light Microscopy (FILM) at Imperial College London is part-supported by funding from the Wellcome Trust (grant 104931/Z/14/Z) and BBSRC (grant BB/L015129/1).

## Author contributions

Conceived the study: C.H.D., designed experiments: C.H.D., K.M.A. and L.L.Y., performed experiments: C.H.D., K.M.A., L.L.Y., R.M. and D.G., analysed data: C.H.D., K.M.A., L.L.Y., J.S., M.H. and M.G., provided tools or reagents: S.R., D.G. and M.H., wrote the manuscript: C.H.D., K.M.A. and L.L.Y. All authors contributed to editing the manuscript.

## Additional information

**Competing interests:** The authors declare no competing interests.

