## [Peer Review File · Nature Communications]

Reviewers' comments:

Reviewer #1 (Remarks to the Author):

In this interesting new paper Akram et al present evidence that epithelial cell migration is required for the appropriate development of alveoli in early postnatal mouse lung. They use an innovative live imaging system, taking advantage of long term culture and advanced deconvolution to provide the clearest videos to date of cell movement during process of alveologenesis. They carefully identify several independent cellular migration behaviors during alveologenesis and go on to demonstrate that inhibition of both microtubule and actin organization both lead to failure of cell migration and reduction of in vitro alveologenesis. Overall, the experiments are carefully conducted and controlled, the manuscript is carefully written and clear, and the model proposed is largely supported by the evidence presented. I think this work provides important pulmonary insights into the mechanisms of the complex process of alveologenesis and will be of significant interest in the developmental biology, lung, and respiratory medicine communities. The technology will be useful in experiments in which mechanisms controlling alveologenesis can be tested. Nonetheless, there are a few issues which if addressed could improve the clarity of the data and the manuscript prior to publication.

1) In Figure 1e, there are several apparently SPC+ but EPCAM- cells present in the imaged field. Are these representatives of imperfect labeling of the epithelium by the EPCAM-FITC reagent in the portions of the cultures imaged? Have the authors assessed the % of epithelial cells properly identified by this reagent? Given the central role of the EPCAM staining in the study, more commentary on this limitation would be helpful for interpretation.

2) In Fig 2 and Fig 7, the authors present proliferation data based on Ki67 staining. The quantitation uses all lung cells as a denominator, but the truly relevant comparison is the amount of proliferative epithelial cells, as the author's model relies on epithelial migration driving alveolar specification and growth. What is the percentage of proliferative epithelial cells observed in their cultures at various times? Figure 7e presents data comparing to SPC, but this is missing for other experimental conditions. How does this percentage compare to in vivo epithelial cell proliferation during alveologenesis? A relevant in vivo comparison (e.g. <https://doi.org/10.1016/j.celrep.2016.11.001>) should be cited.

3) It would be useful to the reader to know the frequency of the observed behaviors. For example, only one instance of cell extension is shown – in how many cultures was this observed? How many alveoli/field in each culture? More discussion of these frequencies would improve interpretation.

4) What is the oxygen tension to which the cultures in PCLS in this study are exposed? How does this differ from physiological oxygen tensions? Alveologenesis in mice is altered by both hyperoxia and hypoxemia with long lasting effects (e.g. [10.1203/PDR.0b013e318211c917](https://doi.org/10.1203/PDR.0b013e318211c917) and <https://doi.org/10.1152/ajplung.00203.2017>) and may impact cell behaviors in vitro as well.

5) The effects of blebbistatin and cytochalasin D in inhibiting cell movement are expected. Since cell toxicity remains an issue regarding these experiments, it would be useful to wash out the drugs and assess restoration of cell movement and processes.

6) Cell movement is known in video 3112037 is interesting, but it is unclear what type of cell is migrating. Staining for epithelial, endothelial cell, myofibroblast and macrophage would be useful to identify the migrating cell.

7) Video 3112038: The authors demonstrate septal like elongation of 2 EPCAM+ cells. Nuclei are not evident in the short ridge – since a number of cells e.g. myofibroblasts, fibroblasts, AT1 and AT2 cells

generates septa. Are multiple nuclei found in some of these “septae”? Do they elongate further?
8) Video 3112043: The authors highlight an area where a small spherical “bleb” like structure appears. How often are these found, do they expand? The identity of the structure shown is not clear from the video.

Reviewer #2 (Remarks to the Author):

The manuscript entitled “Live imaging of alveologenesis reveals dynamic epithelial cell behaviour” investigates mechanisms of alveologenesis by in-depth live-cell imaging techniques of precision cut lung slices PCLS. The authors report of novel three novel cell behaviors (“cell clustering”, “hollowing”, and “cell extension”). The manuscript is very well written, includes interesting data, and applies 300 μm thick PCLS, which provide a great ex-vivo model for the (live) investigation of cellular and even subcellular mechanism. However, I have some serious, including some technical concerns about the presented (mostly descriptive) data and its interpretation thereof, especially in respect of the length of the movies and viability during the live-cell imaging. Here are my details:

(1) The mouse PCLS used in this study were generated by a previous inflation with agarose. Due to our own experience, and which is also discussed in other papers or reviews (e.g.: see Sanderson, M. J. (2011). "Exploring lung physiology in health and disease with lung slices." *Pulm Pharmacol Ther* 24(5): 452-465), the agarose is always present as a plug within the alveoli. Even decellularization does not get rid of the agarose. Without the agarose, all the alveolar space would simply collapse, thus it mimics the positive air space within the alveoli. Anyway, I am asking myself how the agarose would interfere with processes of alveologenesis. Can the authors comment on that and/or include this as a technical discussion point in the discussion part?

(2) In figure 2b of postnatal stage 7, a thickening of the alveolar walls is seen. Is this a normal process during alveolarization or could this also be some sort of artefact of the agarose filling?

(3) The authors use antibodies for their (live) stainings. They should also show some negative controls (e.g. secondary antibody only and/or IgG) as supplementary data.

(4) To me it is unclear, how tracking of the cells was accomplished. Was this done by tracking the nuclei of the cells or by the Epcam staining? Please explain this in more detail in the material and methods.

(5) Overall I see a problem that the authors try to compare tracking data from time-lapse movies of different lengths (e.g. for figure 6 they used 16 hours and in figure they used 12 hours movies). So net-migration distances in μm are not really comparable to each other. For all these measurements the authors should use the same length of movies. In that light, also the migration measurements shown in Figure 2 (here they use 8h movies) are not really comparable to the other migration data (in respect of net cell migration). Also the percentage of cells that migrate $> 4 \mu\text{m}$ will be different depending whether 16h, 12h, or 8h movies are analyzed. For all the migration studies I suggest to use at least 16 hours. Furthermore, a net migration of a maximum of 3 μm (considering that a nucleus has a diameter of 10 μm) is not really much. Is that really directional migration or would the cells just “vibrating” along their fixed position in the tissue? To clarify this, only long term movies of at least 48 hours have to be used.

(6) In the supplementary video 3 B (P7) I see that nearly all Epcam+ cells change their morphology from an elongated to rounded morphology. This might indicate that cells already undergo some sort of cell death during the 8 hours of imaging. Can the authors also see this effect and comment on that?

(7) In the supplementary video 3 A (P3) I can recognize a couple of Epcam+ cells which fuse and form some sphere-like structures (reminiscent of spheroids in other 3D cell culture systems) which start to rotate. Have the authors also observed this and have they investigated this further? Could you please comment on that?

(8) Figure 3 and supplementary video 5: "cell extension": I do not really understand what this observation would tell us in respect of alveolarisation and also in light of that there still might be an agarose plug present. It also might be an immune cells (as stated in this paragraph in the text). It would be crucial to figure out, which kind of cell type it is, e.g.: epithelial cells, neutrophil, macrophage, ?

(9) Figure 4 and supplementary video 6A: I think there is also a longer movie needed in order to see here a clear biological effect. It looks like the beginning of septation, but it does not convincingly show this effect. Probably a magnified view of this ROI would help, in combination with enhancing the overall intensity of this area. Also a colocalization with an α SMA+ myofibroblast, which according to literature, should sit at this position could help.

(10) Figure 4a-b and supplementary video 6B: here I see some focal drift. I think the authors have to show a maximum intensity projection from their z-stacks. Also confocal imaging might help here. However, the effect of cell-clustering just might be an artifact as the focal plane is drifting.

(11) Line 247: "We also observed cell migrating to contribute to an existing alveolus". How can the authors say so? To show an existing alveolus, I think you need 3D/4D data to show the alveolus. Else one just recognizes a sheet of cells. As the Huygens Deconvolution software is used, which reduces blurring and noise from diffraction from wide field images: If the images are "near-confocal" it would enable 3D-reconstruction or at least sequential imaging of layers through the tissue. All in all, in 3D imaging in our hands works better with a confocal system, as the laser light (in combination with bright and stable fluorophores) is gentler to the tissue/cells in respect of photo toxicity. Three dimensional images are needed to conclude morphological changes in a 3D tissue. Focus shifts, tissue movement etc. otherwise lead to significant artefacts, impeding with the interpretation of the seen.

(12) Figure 4a-c and supplementary video 6C, 7: see also comments in (10). I think also here the authors have to show a maximum intensity projection, as again some focal drift is clearly visible.

(13) Assessment of cytotoxicity by MTT assays looks per se good. However, to be on the safe side, I would suggest that the authors also include a live/dead staining directly in the PCLS after the live-cell imaging is done, simply to exclude any cytotoxic/phototoxic effects.

(14) I am curious to know how the authors managed to solve 50 μ M of para-Nitroblebbistatin in medium, as its maximal solubility is reported to be 3.3 μ M after 4 hours in aqueous solution (e.g. compare to para-aminoblebbistatin). Cf. to Varkuti, B. H., et al. (2016). "A highly soluble, non-phototoxic, non-fluorescent blebbistatin derivative." Sci Rep 6: 26141.

Minor:

(1) The exact parameters (Temp, CO₂, which incubator used) during live-cell imaging are not described in the material and methods section.

(2) Typo in Figure S3 (a): should be μ M instead of μ m.

Reviewer #3 (Remarks to the Author):

The topic of alveolar formation is one that has been ignored in lung development, and as the authors stated is extremely important in many chronic and acute lung diseases. Therefore, obtaining direct knowledge of how it occurs thru time-lapse live imaging could add significantly to addressing the pathogenesis of some of these diseases.

The manuscript reports well thought out experiments to elucidate the mechanisms of alveolarization. These experiments are well conducted, and the conclusions are supported by the data. These are the strengths of the study.

The weaknesses are not numerous. However, there is a major weakness in that the study does not substantially add to our present understanding of alveolarization. For example, it is not surprising that cell movement plays a major role in this process. In addition, while the videos are interesting and show good live imaging of the process, they do not help in any substantial way to elucidate the underlying mechanisms. Also, it is difficult to know how such processes as "clustering & hollowing can mediate or promote alveolar formation. Thus, in sum, the studies do not represent new insight or add much new information regarding alveolarization.

Reviewer 1

We thank the reviewer for stating that this work provides important pulmonary insights into the mechanisms of the complex process of alveologenesis and will be of significant interest in the developmental biology, lung and respiratory medicine communities. Also that the technology will be useful in experiments in which mechanisms controlling alveologenesis can be tested.

Answers to specific points

1. In Figure 1e, there are several apparently SPC+ but EPCAM- cells present in the imaged field. Are these representatives of imperfect labelling of the epithelium by the EPCAM-FITC reagent in the portions of the cultures imaged? Have the authors assessed the % of epithelial cells properly identified by this reagent? Given the central role of the EPCAM staining in the study, more commentary on this limitation would be helpful for interpretation.

The reviewer raises an important point, that EpCAM staining is central to our study and therefore the efficiency with which this reagent labels epithelial cells is important to know. We have immunostained P3 PCLS with EpCAM and the epithelial marker antibody pan-cytokeratin and quantified the dual positive cells to show that EpCAM efficiently labels epithelial cells (New Supplemental Fig. 2). We have also the following sentence to the results ‘and the efficiency with which EpCAM labelled epithelial cells was also confirmed by double immunostaining with EpCAM and pan-cytokeratin antibodies (96.98 % of cytokeratin labelled cells were EpCam positive, Figure S2a-d, i)’. We also added some additional references on EpCAM, Hasegawa K. et al. 2017, Fujino N. et al. 2012).

In Fig. 1 panel e, the vast majority of SP-C positive cells are also EpCAM positive. However the pattern of staining for EpCAM and Sp-C within the cells differs as would be expected due to the different localisation of these proteins within the cells. This means that in the panel showing merged staining Fig1 e, first panel, some of the cells don't look dual positive when in fact they are, as seen by comparing the last 2 panels; Epcam only or Sp-C only. However, we do see a small number of SP-C positive cells that are EpCAM -ve in PCLS. We investigated the identity of these cells by immunostaining P3 PCLS for SP-C and the macrophage marker CD11c and we have now added this new data showing that some macrophages contain SP-C (New Supplemental Fig.2 e-h), presumably because they have engulfed some of this secreted protein. This likely accounts for the small number of SP-C positive EpCAM negative cells.

2. In Fig 2 and Fig 7, the authors present proliferation data based on Ki67 staining. The quantitation uses all lung cells as a denominator, but the truly relevant comparison is the amount of proliferative epithelial cells, as the author's model relies on epithelial migration driving alveolar specification and growth. What is the percentage of proliferative epithelial cells observed in their cultures

at various times? Figure 7e presents data comparing to SPC, but this is missing for other experimental conditions. How does this percentage compare to in vivo epithelial cell proliferation during alveologenesis? A relevant in vivo comparison (e.g. <https://doi.org/10.1016/j.celrep.2016.11.001>) should be cited.

We have now added quantification of Ki67 positive and EpCAM/Ki67 dual positive cells in P3 and P7 PCLS to address the reviewer's point that the truly relevant comparison is the amount of proliferative epithelial cells. 'Specific quantification of EpCAM/Ki67 dual positive epithelial cells in PCLS during bulk alveologenesis, showed that in the ex-vivo PCLSi model, the mean total of proliferating cells was 16.1%(P3) and 15.8%(P7) however the level of epithelial cells proliferating accounted for approximately half of this, 9.8%(P3) and 8.8%(P7) (Fig S 5 b). We have also studied the data on proliferation levels in the lungs in vivo and ex vivo in a number of different publications. We found that the methods used, the cell types examined and the levels of proliferation recorded varied. We have altered the introduction to reflect this as follows: 'Cell proliferation is considered to play a key role in alveologenesis, with many publications showing it increases at the onset of bulk alveolarisation around P4 and then rapidly declines towards the end of this developmental phase, however the methods used to measure proliferation and the cell types analysed vary widely between studies, as does the level of proliferation reported (19-22)' We have also added the suggested reference as an in vivo comparison.

Finally, since it is important to discuss the proliferation we see in the PCLS in comparison to other published data on alveolar epithelial cell proliferation, we have added the following text to the discussion: 'Of note, our data revealed that the percentage of epithelial cell proliferation in ex vivo PCLSi, is relatively low during bulk alveologenesis, compared to previous studies in vivo (22)'.

3. It would be useful to the reader to know the frequency of the observed behaviours. For example, only one instance of cell extension is shown – in how many cultures was this observed? How many alveoli/field in each culture? More discussion of these frequencies would improve interpretation.

Thank you for this suggestion. We have quantified the frequencies of the cell behaviours seen in our movies and added the following text on this point: To determine the frequency of septation, cell clustering and hollowing events seen in the movies at P3, we calculated the number of each type of event observed in a total of 12 separate PCLS fields from 3 different imaging experiments. The mean alveolar number per field of view was 5.66, and we observed the following frequencies of cell behaviours per field of view: septation, 1.9; clustering 5.5 and hollowing 1.9.

4) What is the oxygen tension to which the cultures in PCLS in this study are exposed? How does this differ from physiological oxygen tensions? Alveologenesis in mice is altered by both hyperoxia and hypoxemia with long lasting effects (e.g. [10.1203/PDR.0b013e318211c917](https://doi.org/10.1203/PDR.0b013e318211c917) and <https://doi.org/10.1152/ajplung.00203.2017>) and may impact cell behaviours in vitro as well.

In this study, we did not manipulate oxygen levels and instead used the conditions commonly used in cell culture and ex-vivo culture experiments as follows: PCLS were maintained in a heated and humidified chamber with room air oxygen levels (37C, 5% CO₂, approx. 20-21% oxygen). However, as the reviewer indicates, both hyperoxia and hypoxemia influence alveologenesis. Alveolar oxygen tension, the partial pressure of oxygen in the alveoli, is lower than in room air due to factors such as altered water vapour. In our model, PCLS are submerged in fluid and the oxygen levels that diffuse into the media will be lower than in air however, to date we have not measured the levels of oxygen in our media. Given the known effects of altered oxygen levels on alveolar development and in other model systems, e.g. Gomes et al., <https://doi.org/10.1371/journal.pone.0161239>, this is an important area for us to investigate in future. A benefit of the PCLS model is that we can manipulate oxygen levels in the alveoli by changing the level of oxygen in the culture chamber, this is something we will explore in future experiments as the ability to alter oxygen levels is likely to be an additional benefit of the PCLSi model. We have altered the wording in the materials and methods, to make the imaging culture conditions clearer 'The 24-well plate was then transferred to a pre-equilibrated and humidified incubator chamber of an inverted Zeiss Axio Observer widefield epifluorescence microscope and PCLS were kept in the following conditions: 37C, 5% CO₂ and room air oxygen levels, approx. 21%'. We have also added the following wording into the discussion to make this important point about oxygen levels clear. 'In the current study, PCLS were maintained in a heated and humidified chamber with room air oxygen levels during imaging (37C, 5% CO₂, approx. 20-21% oxygen). In future, it would be interesting to investigate the effect of altered oxygen levels on alveologenesis using this technique, given the differential effects demonstrated in other models (37, 38).

5. The effects of blebbistatin and cytochalasin D in inhibiting cell movement are expected. Since cell toxicity remains an issue regarding these experiments, it would be useful to wash out the drugs and assess restoration of cell movement and processes.

We appreciate that suggestion to conduct cultures with Cytochalasin D and Blebbistatin where the drugs are washed out and then restoration of cell movement is assessed. However, in our experiments we left the inhibitors on for the duration of the experiments as it is technically challenging to carry out multiple media changes needed to wash out the drug whilst they are in the imaging set up. Despite the presence of drugs for the duration of the experiment, we show that there is no significant difference in viability of PCLS cultured with or without inhibitors at the end of the imaging period (Fig6j, Fig8f).

6) Cell movement is known in video 3112037 is interesting, but it is unclear what type of cell is migrating. Staining for epithelial, endothelial cell, myofibroblast and macrophage would be useful to identify the migrating cell.

We have included a new zoomed in movie showing another cell extension event, this time from an EpCAM labelled PCLS (Supplemental video 11 Cii). In this movie an EpCAM positive rapidly extending cell can be seen elongating around an alveolar wall.

We have added movies showing endothelial cells to the manuscript (Supplemental videos 12 and 13) but we do not see this type of extension event with endothelial cells. We also attempted to conduct movies by labelling PCLS with CD140 α but we were unable to obtain efficient labelling. However, given that we have identified an EpCAM positive extending cell we believe that these rapidly elongating cells are epithelial. It is possible that this type of event could be an epithelial cell adopting a type I cell morphology. We have added the following text on this to the discussion: 'Our identification of an EpCAM positive cell extending round an alveolar wall raises the interesting possibility that these elongating cells could be epithelial cells differentiating into type I alveolar cells but further studies will be required to test this hypothesis'

7) Video 3112038: The authors demonstrate septal like elongation of 2 EPCAM+ cells. Nuclei are not evident in the short ridge – since a number of cells e.g. myofibroblasts, fibroblasts, AT1 and AT2 cells generates septa. Are multiple nuclei found in some of these “septae”? Do they elongate further?

We have included new zoomed-in movies of the septation events that we highlighted in supplemental 5A (old supplemental movie 6a). In these 3D movies, septation can be seen more clearly and it is possible to see the septal elongations extending further into the space Supplemental movies 5B and 5Ci. Almost complete septation is visible in movie 5Ci where one green projection from the bottom of the field joins with a projection from the top of the field to sub-divide the space. Whilst we have not carried out movies with multiple cell types labelled altogether, to minimise phototoxicity, in Supplemental movie 5B in particular, multiple nuclei can be seen at the point where the projection arises, several of these cells are not EpCam positive but the projections are EpCAM positive indicating that they are epithelial cells. In addition, we have also added movies of PCLS labelled with an endothelial cell marker and from these, it can be seen that capillaries are present in the extending septa alongside the epithelial cells (supplementary movies 12 and 13).

8) Video 3112043: The authors highlight an area where a small spherical “bleb” like structure appears. How often are these found, do they expand? The identity of the structure shown is not clear from the video.

We observed hollowing 23 times within 12 separate PCLS movie fields, a frequency that was similar to that of septation. The small holes that appear within the interstitium such as that shown in Supplemental video 7 (old video 9) do expand over time (see Fig. 4a, showing the hollow at 0, 9 and 18hr). We have measured the diameter and added this into the results 'the hole increased in diameter during the imaging period from 0 to 3.8 μ m at 9 hours and 8.5 μ m at 19 hours'. We have also quantified the increase in volume during other hollowing

events shown in Figure 4B and new Supplementary video 8, white circles). 'Area A expands from a volume of $32.55\mu\text{m}^2$ at 0 hours to $778.98\mu\text{m}^2$ at 14 hours and area B from $2.55\mu\text{m}^2$ at 0 hours to $160.76\mu\text{m}^2$ at 14 hours'.

Reviewer 2

1) The mouse PCLS used in this study were generated by a previous inflation with agarose. Due to our own experience, and which is also discussed in other papers or reviews (e.g.: see Sanderson, M. J. (2011). "Exploring lung physiology in health and disease with lung slices." *Pulm Pharmacol Ther* 24(5): 452-465), the agarose is always present as a plug within the alveoli. Even decellularization does not get rid of the agarose. Without the agarose, all the alveolar space would simply collapse, thus it mimics the positive air space within the alveoli. Anyway, I am asking myself how the agarose would interfere with processes of alveologenesis. Can the authors comment on that and/or include this as a technical discussion point in the discussion part?

We have added the following text into the discussion on this important point. 'We confirmed that despite the likelihood that at least some agarose is retained in the airspaces of PCLS (23), alveologenesis is able to occur, in agreement with a previous study of alveologenesis in static PCLS (19). The presence of agarose in PCLS, prevents the alveoli from collapsing and in this way mimics the in vivo alveolar environment'.

2) In figure 2b of postnatal stage 7, a thickening of the alveolar walls is seen. Is this a normal process during alveolarization or could this also be some sort of artefact of the agarose filling?

We also noticed that in postnatal day 7 the lung slices have relatively thicker alveolar wall than P3, P14 and adult lung slices. However, this is something we consistently observe in many different P7 lungs both in this and previous studies (Poobalasingam et al. *Dis. Model Mech* 2017 10:409-423). We think this is a normal feature of the lung architecture at this age rather than an artefact from filling the lungs with agarose.

3) The authors use antibodies for their (live) stainings. They should also show some negative controls (e.g. secondary antibody only and/or IgG) as supplementary data.

We apologise that these important controls were not included in our original manuscript. We have now included IgG controls for all the fluorophores used in used in our live imaging experiments in new Supplemental Figure 8.

4) To me it is unclear, how tracking of the cells was accomplished. Was this done by tracking the nuclei of the cells or by the Epcam staining? Please explain this in more detail in the material and methods.

Cell tracking was done using the EpCAM staining and only EpCAM-positive epithelial cells were tracked. We have added further details on the cell tracking

methods used into the materials and methods. In the revised manuscript, we have also included a different set of cell tracking data to provide the reader with a better overview of the extent to which cells were migrating in the PCLS movies. We have replaced the graphs that showed cell migration of greater than 4 μ m, with new graphs showing individual net cell migration, which reflect the actual migration of each cell; these graphs show the distribution of cells migrating between 0 and 14 μ m (New figures: Fig 2j, Fig 6n, Fig 8e).

5) Overall I see a problem that the authors try to compare tracking data from time-lapse movies of different lengths (e.g. for figure 6 they used 16 hours and in figure they used 12 hours movies). So net-migration distances in μ m are not really comparable to each other. For all these measurements the authors should use the same length of movies. In that light, also the migration measurements shown in Figure2 (here they use 8h movies) are not really comparable to the other migration data (in respect of net cell migration). Also the percentage of cells that migrate > 4 μ m will be different depending whether 16h, 12h, or 8h movies are analyzed. For all the migration studies I suggest to use at least 16 hours. Furthermore, a net migration of a maximum of 3 μ m (considering that a nucleus has a diameter of 10 μ m) is not really much. Is that really directional migration or would the cells just “vibrating” along their fixed position in the tissue? To clarify this, only long term movies of at least 48 hours have to be used.

In our manuscript we have only compared cell tracking data between groups where movie analysis and cell tracking data is of the same duration. In Fig2, we have used 8hr duration as we wished only to determine the age at which cell migration was optimal. It is not possible to directly compare the data from Fig. 2 to that from Fig.6 and 8 because in Fig. 6 and 8 DMSO was present in the media and we have not made any comparisons between these data sets.

However, taking the reviewers suggestion, we have gone back to the raw data and generated new movies and cell tracking of 14 hours duration for both blebbistatin (Fig 6 and 7) and Cytochalasin D (Fig. 8) treatments. There was some bleaching of the EpCAM signal in the at the end of the Cytochalasin D movies (beyond 14 hours) and therefore we only analysed all data up to 14 hours for both Blebbistatin and Cytochalasin D.

As the reviewer suggests, it would be excellent to conduct movies of alveogenesis for 48 hours in order to visualise the process over a longer time period. However, we found that the PCLS were not viable beyond around 20 hours of culture, see results P5 ‘Longer time-lapse experiments (up to 64 hours) were also conducted, however, the tissue was not viable after approximately 21-24 hours, compared to PCLS cultured under normal (non-time-lapse) conditions and therefore longer-term experiments were discontinued (Supplementary video 1A, B; Figure S1e)’. Furthermore, the FITC flurophore signal became significantly bleached beyond 19 hours, see results P5 ‘Both of these live cell labels allowed individual cells to be tracked in the PCLS for up to 19 hours without significant bleaching and did not alter cell viability during time-lapse imaging (Figure 1d)’. It was therefore not possible to conduct movies of longer duration.

We also deliberately chose to use widefield microscopy to reduce fluorophore bleaching, see results p6 'To reduce fluorophore bleaching during time-lapse acquisition, we opted to use widefield microscopy (31).'

It is however highly novel to successfully conduct such long-term imaging of live tissue showing individual cells as we have highlighted in our discussion: 'Use of this technique to image thick tissue samples and for such a long time period is highly novel. Importantly, the combination of cell labelling with widefield microscopy and deconvolution avoids the need for expensive confocal microscopes...'

6) In the supplementary video 3 B (P7) I see that nearly all Epcam+ cells change their morphology from an elongated to rounded morphology. This might indicate that cells already undergo some sort of cell death during the 8 hours of imaging. Can the authors also see this effect and comment on that?

In the original P7 movie, the EpCAM staining was a little overexposed, making it difficult to decipher the outlines of each individual cell, particularly at the beginning of the movie. We have replaced the original movie of P7 PCLS with a different one in which the EpCAM is less saturated. In this movie Although we consistently find that EpCAM staining is particularly strong at P7, the outline of cells can be seen more clearly. In this movie rounded cells can be seen throughout the movie, similar to those at P3. We confirmed that cells are viable at the end of imaging by live/dead staining of P3 PCLS, where many cells of rounded morphology are visible in the movies.

7) In the supplementary video 3 A (P3) I can recognize a couple of Epcam+ cells which fuse and form some sphere-like structures (reminiscent of spheroids in other 3D cell culture systems) which start to rotate. Have the authors also observed this and have they investigated this further? Could you please comment on that?

We had noticed these spheroid-like groups of cells that rotate in some of our movies. However, most frequently we see clustering of greater numbers of cells together e.g. as seen in Supplemental movie 5A and 5D. We have not investigated these structures further at the moment but I agree that it is reminiscent of spheroids in other 3D culture systems and I suspect it has something to do with the cells adjusting their polarity to be uniform as can be seen in sections through organoids e.g. Choi J, Iich, E., Lee J-H 2016 Dev Biol. 420:278-286.

8) Figure 3 and supplementary video 5: "cell extension": I do not really understand what this observation would tell us in respect of alveolarisation and also in light of that there still might be an agarose plug present. It also might be an immune cells (as stated in this paragraph in the text). It would be crucial to figure out, which kind of cell type it is, e.g.: epithelial cells, neutrophil, macrophage, ?

We have included a new zoomed in movie showing another cell extension event, this time from an EpCAM labelled PCLS (Supplemental video 11 Cii). In this movie an EpCAM positive rapidly extending cell can be seen elongating around an alveolar wall. We have added movies showing endothelial cells to the manuscript (Supplemental videos 12 and 13) but we do not see this type of extension event with endothelial cells. We also attempted to conduct movies by

labelling PCLS with CD140 α but we were unable to obtain efficient labelling. However, given that we have identified an EpCAM positive extending cell we believe that these rapidly elongating cells are epithelial. It is possible that this type of event could be an epithelial cell adopting a type I cell morphology. We have added the following text on this to the discussion: 'Our identification of an EpCAM positive cell extending round an alveolar wall raises the interesting possibility that these elongating cells could be epithelial cells differentiating into type I alveolar cells but further studies will be required to test this hypothesis'

9) Figure 4 and supplementary video 6A: I think there is also a longer movie needed in order to see here a clear biological effect. It looks like the beginning of septation, but it does not convincingly show this effect. Probably a magnified view of this ROI would help, in combination with enhancing the overall intensity of this area. Also a colocalization with an α SMA+ myofibroblast, which according to literature, should sit at this position could help.

To more clearly show the septation events in Figure 4 and Supplementary video 6A we have taken the suggestion to obtain 3D movies of the regions of interest where septation is occurring. These new movies, Supplementary movie 5B and 5C show septation more clearly. In addition, the cells protruding into the space are EpCAM positive, indicating that they are epithelial cells. However there are clearly other cell nuclei present at these points (Movie 5B), which are likely to be non-epithelial cells that also contribute to septation, as the reviewer points out. Please also see response to reviewer 1 point 7 for further information on this point.

10) Figure 4a-b and supplementary video 6B: here I see some focal drift. I think the authors have to show a maximum intensity projection from their z-stacks. Also confocal imaging might help here. However, the effect of cell-clustering just might be an artifact as the focal plane is drifting.

The new 3D movies we have added (supplemental movies 5A and 5D more clearly show areas of cell clustering in a P3 PCLS and that this is a specific cell behaviour in alveologenesis rather than an artefact. We have also added data to the results about the frequency of cell clustering in our PCLS movies 'we observed the following frequencies of cell behaviours per field of view: septation, 1.9; clustering 5.5 and hollowing 1.9'.

(11) Line 247: "We also observed cell migrating to contribute to an existing alveolus". How can the authors say so? To show an existing alveolus, I think you need 3D/4D data to show the alveolus. Else one just recognizes a sheet of cells. As the Huygens Deconvolution software is used, which reduces blurring and noise from diffraction from wide field images: If the images are "near-confocal" it would enable 3D-reconstruction or at least sequential imaging of layers through the tissue. All in all, in 3D imaging in our hands works better with a confocal system, as the laser light (in combination with bright and stable fluorophores) is gentler to the tissue/cells in respect of photo toxicity. Three dimensional images are needed to conclude morphological changes in a 3D tissue. Focus shifts, tissue movement etc. otherwise lead to significant artefacts, impeding with the interpretation of the seen.

We have taken the reviewers suggestion and included new 3D movies of alveologenesis. New supplemental movie 5E shows a 3D zoomed in view of a5 and a6 from movie 5A. Here it can be seen that epithelial cells migrate and integrate with other epithelial cells forming a wall surrounding two airspaces that are 3 dimensional.

(12) Figure 4a-c and supplementary video 6C, 7: see also comments in (10). I think also here the authors have to show a maximum intensity projection, as again some focal drift is clearly visible.

We have replaced these movies with new 3D movies. See answers to point 9, 10 and 11 above.

(13) Assessment of cytotoxicity by MTT assays looks per se good. However, to be on the safe side, I would suggest that the authors also include a live/dead staining directly in the PCLS after the live-cell imaging is done, simply to exclude any cytotoxic/phototoxic effects.

We have taken the authors suggestion and included a new supplemental figure showing live/dead staining on 3 individual P3 PCLS at t=0 and at the end of 16 hours of imaging (New Figure S4). This confirms that the slices are still viable after imaging.

14) I am curious to know how the authors managed to solve 50uM of para-Nitroblebbistatin in medium, as its maximal solubility is reported to be 3.3uM after 4 hours in aqueous solution (e.g. compare to para-aminoblebbistatin). Cf. to Varkuti, B. H., et al. (2016). "A highly soluble, non-phototoxic, non-fluorescent blebbistatin derivative." Sci Rep 6: 26141.

The nitro-Blebbistatin we used in our experiments was first solubilised in an organic solvent as follows: we dissolved the nitro-Blebbistatin in DMSO to a concentration of 5mg/ml and then diluted into the aqueous media from this stock to a final concentration of 50uM.

Reviewer 3

We thank the reviewer for their comments:

The topic of alveolar formation is one that has been ignored in lung development, and as the authors stated is extremely important in many chronic and acute lung diseases. Therefore, obtaining direct knowledge of how it occurs thru time-lapse live imaging could add significantly to addressing the pathogenesis of some of these diseases. The manuscript reports well thought out experiments to elucidate the mechanisms of alveolarization. These experiments are well conducted, and the conclusions are supported by the data. These are the strengths of the study. The weaknesses are not numerous. However, there is a major weakness in that the study does not substantially add to our present understanding of alveolarization. For example, it is not surprising that cell movement plays a major role in this process. In addition, while the videos are interesting and show good live imaging of the process, they do not help in any substantial way to

elucidate the underlying mechanisms. Also, it is difficult to know how such processes as “clustering & hollowing can mediate or promote alveolar formation. Thus, in sum, the studies do not represent new insight or add much new information regarding alveolarization.

We have addressed the reviewers specific concerns below:

The data presented in our manuscript provides important contributions to the field of respiratory biology in two ways. First it shows novel aspects of cell biology that contribute to alveologenesis and second, the PCLSi model that we have established is the first and currently the only *ex vivo* model that can be used to study alveologenesis in real time.

Current dogma in the field is that alveoli form by repeated septation events that sub-divide the alveolar walls thereby providing an ever-greater surface area to support gas-exchange. In this manuscript, we have been able to show septation events happening live for the first time. In addition to this, we have focused predominantly on cellular mechanisms that contribute to alveologenesis by establishing live imaging of this process. We show for the first time that in addition to septation, alveoli can form using alternative strategies. We have discovered the following strategies that contribute to alveolarisation:

- 1) Cell clustering, where groups of cells migrate and coalesce (Figure 2a,b and movie 5A and D).
- 2) Hollowing where a hole forms within a cluster of cells. We see both formation of new holes where one did not previously exist (Figure 3a and movie 7) and widening of existing alveolar spaces to form larger airspaces (Figure 3b and movie 8 and 9).
- 3) In alveoli where a distinct airspace exists, we also see epithelial cells migrating and integrating into the existing alveolar walls to contribute to the existing alveolus (Figure 2c and movie 5a and 5E).

We have discussed our findings with respect to these novel aspects of alveologenesis as follows:

‘We identified several novel cell behaviours that contribute to alveologenesis. Based on careful analysis of all our movies, we propose that these behaviours represent different stages or strategies for alveolar development from *de novo* formation to terminal maturation. Cell clustering, where epithelial cells migrate to form a tight cluster of cells in one area, appears to represent an early stage in alveolar development (Figure 8j A) and this was the most frequent cell behaviour seen in our movies. Hollowing initially occurs within a cell cluster; at the beginning of this process the cluster of epithelial cells rearranges to enable a new hole to open up (Figure 8j B,ii). Subsequently, further hollowing occurs to establish a 3-dimensional pocket, this includes broadening of the initial small hole to form an airspace (Figure 8j B,iii) Once an alveolus is formed, further epithelial cells can migrate towards this and integrate into the nascent alveolar wall, thereby contributing to the mature structure (Figure 8j B,iv). We frequently observed epithelial cells migrating towards existing airspaces and contacting another cell already around the airspace.’

From analysing our movies, we conclude that the novel behaviours exist alongside septation as mechanisms to form alveoli.

4) Finally, we have discovered cells that rapidly spread and extend around the wall of an existing alveolar airspace (Figure 4 and movie 10). We have named this 'cell extension'. We have shown that these cells are EpCAM positive, indicating that they are epithelial cells (movie 11 Cii). We speculate in our discussion that these could be type II cells differentiating into type I cells.

We have included the following text on this point in our discussion:

'Our identification of an EpCAM positive cell extending round an alveolar wall raises the interesting possibility that these elongating cells could be epithelial cells differentiating into type I alveolar cells but further studies will be required to test this hypothesis'.

The discovery of these aspects of epithelial cell behaviour underline the importance of being able to visualise alveologenesis in real-time in order to provide a holistic understanding of this important process. The breadth of information gained from live imaging of this process can be likened to the information gained from being a spectator at a live football game compared to looking at a single photograph used to represent the match in a newspaper.

In addition to discovering novel strategies that contribute to alveologenesis, our manuscript shows the PCLSi model is a valuable tool that can be used to investigate the effect of interventions, such as the cytoskeleton inhibitors used in our manuscript, on the process of alveologenesis. We have outlined further advantages of this model system in the discussion: 'In future, this technique could be used for more detailed investigation of the capillary endothelium and to investigate the role of other cell populations such as fibroblasts in alveologenesis. Moreover, the PCLSi technique could be used with tissue from genetically modified mice to investigate genetic and cellular contributions to alveologenesis. This technique could also be adapted for tissue slices from other organs and species, including humans'. There are currently no other ex vivo model systems that can be used to study alveologenesis in real time. Furthermore, this novel model system may be used to develop potential regenerative treatments for the many currently incurable lung diseases that result from the failure of distal lung repair, including: COPD, fibrosing lung diseases and acute respiratory distress syndrome.

REVIEWERS' COMMENTS:

Reviewer #1 (Remarks to the Author):

The authors have addressed the major issues raised by all of the reviewers. The work provides new methods that will enable new mechanistic studies regarding an important area of lung biology. While the research community may expect cell extension, directional movements and proliferation to be involved in the process of septation and alveolarization, the present this work provides real data demonstrating cell clustering, hollowing and extension of epithelial cells, providing the basis for exploring signal-transduction pathways by which alveolarization occurs. The inhibitor studies help verify that cell movement, dependent on actin-myosin and are not random or artifactual movements. While descriptive, the work provides the framework for study of epithelial cell processes and their interactions with matrix and other cells that can now be studied in more mechanistic experiments. As presented, the work is useful for the field.

Reviewer #2 (Remarks to the Author):

My major concerns have been addressed. The quality of the images and videos has substantially improved. It is impressive to see an onset of alveolar septation live. Pity, that the movie is not taking longer due to the reduced viability of the PCLS.